# Ligand recognition and activation of neuromedin U receptor 2

Wenli Zhao[1,2,3,9], Wenru Zhang[4,9], Mu Wang[1,2,5,9], Minmin Lu[1,2,3,9], Shutian Chen[1,2,3], Tingting Tang[1,2,3], Gisela Schnapp [6], Holger Wagner[6], Albert Brennauer[6], Cuiying Yi[1,2], Xiaojing Chu[1,2], Shuo Han [1,2,7] ✉, Beili Wu [2,3,7] ✉ & Qiang Zhao [1,3,4,8] ✉

Neuromedin U receptor 2 (NMU2), an emerging attractive target for treating obesity, has shown the capability in reducing food intake and regulating energy metabolism when activated. However, drug development of NMU2 was deferred partially due to the lack of structural information. Here, we present the cryo-electron microscopy (cryo-EM) structure of NMU2 bound to the endogenous agonist NmU-25 and $G_{i1}$ at 3.3 Å resolution. Combined with functional and computational data, the structure reveals the key factors that govern the recognition and selectivity of peptide agonist as well as non-peptide antagonist, providing the structural basis for design of novel and highly selective drugs targeting NMU2. In addition, a 25-degree rotation of $G_i$ protein in reference to NMU2 is also observed compared in other structures of class A GPCR−$G_i$ complexes, suggesting heterogeneity in the processes of G protein-coupled receptors (GPCRs) activation and G protein coupling.

Neuromedin peptides, a structurally and functionally diverse neuropeptide family, consist of four groups of related peptides, the bombesin-like peptides (NmB and NmC), the kassinin-like peptides (NmK and NmL), a neurotensin-like peptide (NmN) and the neuromedin U group (NmU and NmS)[1,2]. Unlike other neuromedins, NmU peptides from different species share an identical C-terminal pentapeptide with a conserved C-terminal amidation[3-5]. There are two subtypes of structurally related NmU peptides, NmU-25 and NmS-33, in humans. NmU-25 is ubiquitously distributed in the gastrointestinal tract, spinal cord, and central nervous system[6,7], while NmS-33 shows a more restricted distribution, predominantly expressed in the central nervous system[8]. The system of NmU-25/NmS-33 implicates in the regulation of smooth muscle contraction, energy balance, feeding behavior, pronociceptive, and tumorigenesis[7,8]. By activating two

GPCRs, NMU1 and NMU2, NmU peptides are related to multiple pathophysiological roles in diabetes, metabolic disorder, inflammation, and cancer[3,8,9]. NMU1 and NMU2 share almost 40% sequence identity and recognize NmU-25 and NmS-33 with high affinity[4]. NMU1 is mainly found in the periphery tissues with the function of regulating intestinal motility and smooth muscle contraction, whereas NMU2 is predominantly expressed in the central nervous system and elicits a response to food intake and nociception[10,11].

To discover the biological functions of NmU peptides and develop practical drug candidates, several NmU peptide analogs with metabolic stability and NMU-subtype selectivity have been developed[12-14]. One selective small molecular antagonist, R-PSOP, was identified with high potency and selectivity for NMU2 ($K_i = 52$ nM) and served as a tool for further exploring the biological roles of NMU2. It has been shown

[1]State Key Laboratory of Drug Research, Shanghai Institute of Materia Medica, Chinese Academy of Sciences, 555 Zuchongzhi Road, Pudong, Shanghai 201203, China. [2]CAS Key Laboratory of Receptor Research, Shanghai Institute of Materia Medica, Chinese Academy of Sciences, Shanghai 201203, China. [3]University of Chinese Academy of Sciences, No. 19A Yuquan Road, Beijing 100049, China. [4]School of Chinese Materia Medica, Nanjing University of Chinese Medicine, Qixia District, Nanjing 210023, China. [5]School of Life Science and Technology, ShanghaiTech University, 393 Hua Xia Zhong Road, Pudong, Shanghai 201210, China. [6]Boehringer-Ingelheim Pharma GmbH & Co. KG, Department of Medicinal Chemistry, Birkendorfer Str. 65, 88397 Biberach, Germany. [7]School of Pharmaceutical Science and Technology, Hangzhou Institute for Advanced Study, UCAS, Hangzhou, China. [8]Zhongshan Branch, the Institute of Drug Discovery and Development, CAS, Zhongshan, China. [9]These authors contributed equally: Wenli Zhao, Wenru Zhang, Mu Wang, Minmin Lu. ✉e-mail: hanshuo10@simm.ac.cn; beiliwu@simm.ac.cn; zhaoq@simm.ac.cn

that activation of NMU2 dramatically decreases body weight and food intake in mice while, in contrast, inhibition of NMU2 promotes weight gain and aspiration for obesogenic food[15-18]. Unfortunately, drug development targeting NMU2 has been limited, partially due to the lack of structural information. To reveal the molecular details of ligand recognition and subtype-selectivity of NMUs, we report the cryo-EM structure of the NMU2–$G_{i1}$ complex bound to the endogenous peptide NmU-25. Together with mutagenesis and molecular docking studies, the structure represents the key signature shared by NmU peptides which is a prerequisite for ligand recognition as well as the mechanism of ligand selectivity. Moreover, we also capture a specific G protein-coupling conformation of NMU2, providing a frame image for the variable activation process between GPCRs and G protein.

## Results

### Cryo-EM structure of NmU-25–NMU2–$G_{i1}$ complex

To facilitate expression and purification of the NmU-25–NMU2–$G_{i1}$ complex, a hemagglutinin (HA) signal peptide followed by a flag tag was introduced at the N terminus of NMU2, while the flexible C-terminal residues of the receptor (Q356-T415) were replaced by a PreScission protease site followed by a twin-strep affinity tag (Supplementary Fig. 1a, b). The addition of tags and deletion of flexible terminus have little effect on NmU-25-induced receptor signaling as indicated by bioluminescence resonance energy transfer $G\alpha_{i1}\beta\gamma$ biosensor (TRUPATH) where the engineered construct showed a similar $pEC_{50}$ value compared to the wild-type (WT) receptor (Supplementary Fig. 1c, d)[19]. Dominant-negative $G\alpha_{i1}$ ($DNG\alpha_{i1}$) containing five mutations (S47C, G202T, G203A, E245A, and A326S) was used to improve the stability of the complex as these mutations lead to a preference for a nucleotide-free state, and prevent the dissociation of Gβγ from the heterotrimer[20,21]. Over 8700 movies were collected, and the structure of the NmU-25–NMU2–$G_{i1}$ complex was determined by cryo-EM single-particle analysis at a global nominal resolution of 3.3 Å (Supplementary Fig. 2a–f, Supplementary Table 1). Each component can be modeled unambiguously with a clear and strong density map (Supplementary Fig. 2g).

The overall structure of NMU2 possesses a canonical seven transmembrane helical domain similar to other solved peptide receptors of class A GPCRs with helix VIII unmodelled due to its flexibility upon activation (Fig. 1a)[22,23]. Two short antiparallel β-strands are formed in extracellular loop 2 (ECL2) and stabilized through a conserved disulfide bond between C119[3.25] and C204[ECL2] (superscript numbers represent Ballesteros–Weinstein nomenclature[24]). To accommodate the peptide ligand, the extracellular part of NMU2 is more widely opened compared with other solved peptide-bound receptors, which share a high sequence similarity with NMU2 (Fig. 1b)[25,26]. Compared with the inactive structure of $NTS_1$, NMU2 adopts a fully active conformation on the intracellular side that is stabilized by G protein coupling with a remarkable outward movement of helix VI (-10 Å, measured by $C_\alpha$ of R[6.30]) as well as the inward movement of helix VII (~4.5 Å, measured by $C_\alpha$ of L[7.55]) (Fig. 1c)[25,27]. Besides the movement of transmembrane helices, many structural features also indicate that NMU2 is in an active state. W281[6.48], which is termed as "toggle switch" and significant for GPCR activation, is in an active-like conformation and induces a rotamer switch of F277[6.44] as well as the rearrangement of the "$P^{5.50}$-$I^{3.40}$-$F^{6.44}$" motif and initiates the outward movement of helices V and VI (Supplementary Fig. 3a–d)[28]. Additionally, the disruption of the helices II–III–VII network leads to the collapse of the interactions between helices III and VII, causing the rearrangement of the "NPxxY" motif (Supplementary Fig. 3e, f). To allow insertion of the α5 helix of G protein, R144[3.50] is released to form a hydrogen network with Y236[5.58] and Y327[7.53] to create a cavity for G protein coupling (Supplementary Fig. 3g, h). All these residues share highly conserved sequence identity, indicating a conserved activation mechanism between NMUs (Supplementary Fig. 4).

### NmU-25 recognition at NMU2

Among mammal species, NmU peptides with various lengths share the conserved heptapeptide at the C terminus (-F-L-F-R-P-R-N-NH₂), including the amidation of the last asparagine (Fig. 2a), which is crucial for the ligand's agonistic activity[29]. NmU-25, the 25 amino acid endogenous agonist, forms a short β-meander structure and the conserved peptide C terminus penetrates deeply in the entire orthostatic ligand binding pocket formed by ECL2, ECL3, and helices I–IV as well as helices VI–VII of NMU2 (Fig. 2b). Unlike the significant roles of helix V ($D^{3.32}$–$S^{5.42}$–$S^{5.46}$ motif) of monoamine receptors in maintaining ligand

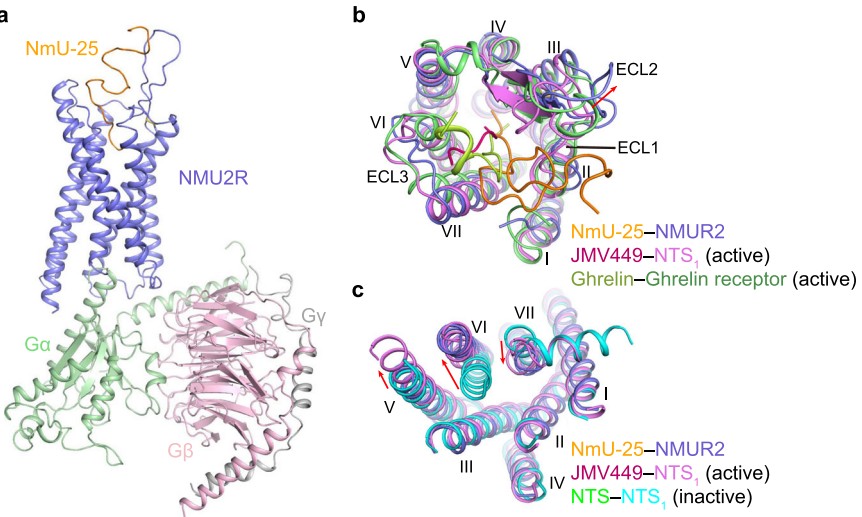

**Fig. 1 | Overall structure of the NmU-25–NMU2–$G_{i1}$ complex. a** Cryo-EM structure of the NmU-25–NMU2–$G_{i1}$ complex is shown in cartoon representation. NmU-25, NMU2, Gα, Gβ, and Gγ are colored orange, slate, light green, light pink, and gray, respectively. The disulfide bond is shown as yellow sticks. **b** Comparison of the ligand-binding pocket in the structure of NmU-25–NMU2 and other peptide-bound receptors. $NTS_1$ (PDB ID: 6OS9) and ghrelin receptor (PDB ID: 7F9Y) are represented as violet and lime cartoon. Ligands are shown as hot pink and limon cartoons. The red arrow indicates the movement of the ECL2 in the NmU-25-bound NMU2 structure. **c** Comparison of the transmembrane helical bundle conformation in the structure of NMU2–$G_{i1}$ and $NTS_1$ (PDB ID: 4BUO, 6OS9). Structures of $NTS_1$ are represented by cyan and violet cartoons, respectively. The red arrows indicate the movement of the intracellular tips of helices V, VI, and VII.

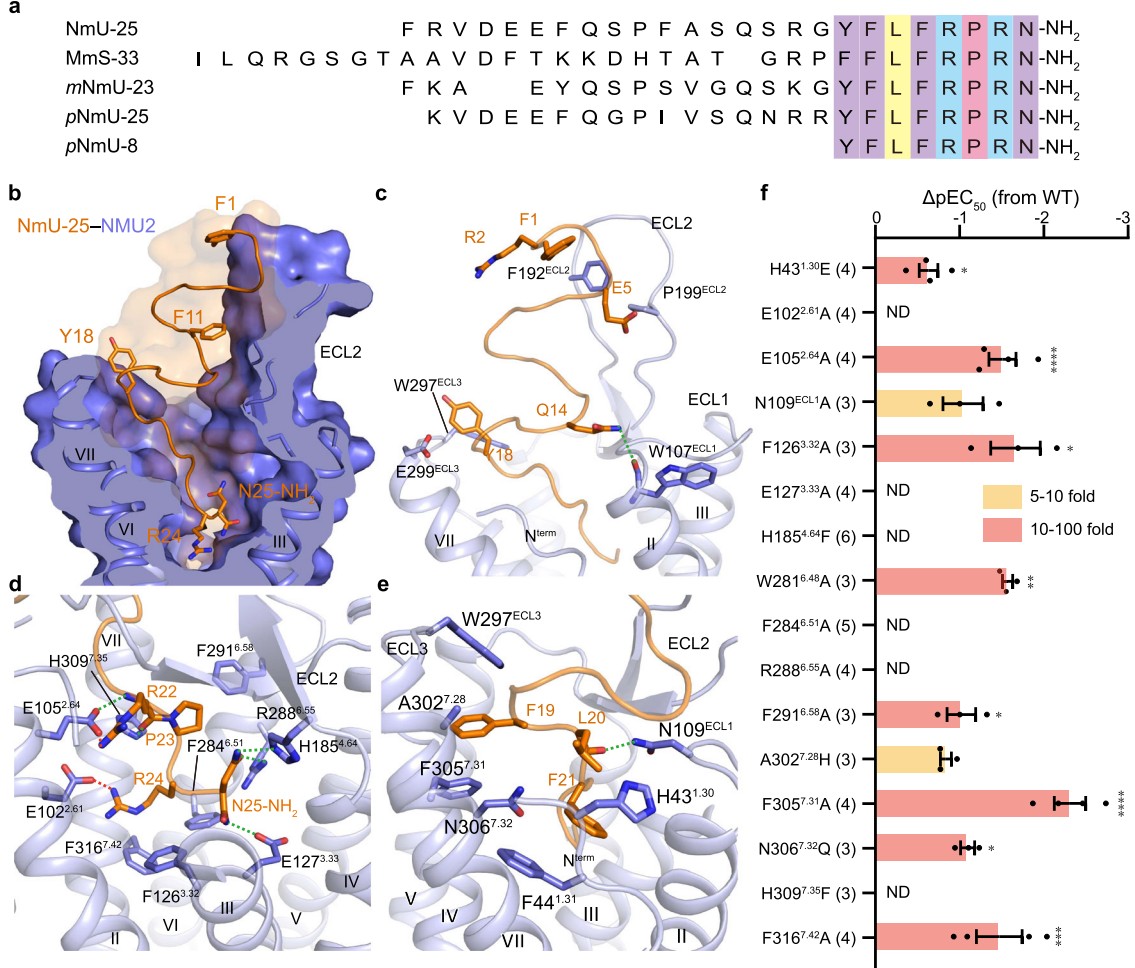

**Fig. 2 | Overall NmU-25 binding mode in NMU2. a** Sequence alignment of NmU peptides from different species. Alignment was carried out by align tool in the UniProt database (https://www.uniprot.org/). The UniProt ID of NmU-25, NmS-33, *m*NmU-23 and *p*NmU-25/8 are P48645, Q5H8A3, Q9QXK8, and P34964, respectively. The conserved C-terminal heptapeptide is indicated by colorful square frames. **b** The cutting face of the ligand binding pocket in the NmU-25–bound NMU2 structure. NmU-25 is represented as orange sticks and cartoons with transparent surfaces. NMU2 is shown as a slate cartoon with a light blue surface. **c** Interactions between the N-terminal segment of NmU-25 and the NMU2. The residues of the receptor and the ligand are shown as slate and orange sticks. A hydrogen bond is indicated by a green dash line. **d** Close view of the interactions between the conserved C terminus of NmU-25 and the NMU2. Salt bridges are indicated by red dash lines. **e** Detailed interactions between the F19-F21 of NmU-25 and the NMU2. **f** NmU-25–induced G protein activation of NMU2 mutants tested by TRUPATH. Bars represent the differences between the calculated NmU-25 potency (pEC$_{50}$) of each mutant relative to WT. Data are shown as mean ± SEM (bars) from at least three independent experiments with individual data points shown (dots). The number of independent experiments is shown in parentheses. One-way ANOVA was performed, followed by Dunnett's post-test, and compared with WT. The $P$ value was defined as: *$P < 0.05$; **$P < 0.01$; ***$P < 0.001$; ****$P < 0.0001$. Supplementary Table 2 provides a detailed statistical evaluation, $P$ values, and expression levels. Source data are provided as a Source Data file.

binding and receptor activation, helix V of NMU2 plays a minor role in ligand-mediated functions (Supplementary Fig. 5)[30]. The receptor and peptide ligand form an interaction area of 1214 Å², and the C terminus of NmU-25 (form F19 to N25-NH$_2$) contributes a large portion of the interaction area (930 Å²), while the N terminus (form F1 to Y18) is accompanied by ECL2 winding alone to the extracellular milieu, leading to the outward movement of ECL2 compared with NTS$_1$ (Fig. 1b). The structure of F1–Y18 forms three kinks to facilitate the accommodation in the space between ECL1–ECL3 and forms limited interactions with the receptor (Fig. 2c). For example, Y18 packs against the main chain of ECL3 by hydrophobic interactions, while Q14 is anchored by a hydrogen bond with W107$^{ECL1}$. In addition, P199$^{ECL2}$ forms hydrophobic interactions with the main chain of E5 to stabilize the N-terminal structure of NmU-25 (Fig. 2c). There has been some indication that NmS-33 may have a higher binding affinity for NMU2 compared to NmU-25[8]. This might result from potentially more interactions between the hydrophilic residues, such as arginine or glutamine in the extended N terminus of NmS-33 and ECL2 of the receptor,

providing a strengthened interaction network between the N terminus of NmS-33 and the receptor. However, more data are needed before a clear conclusion can be made.

The last four C-terminal residues of NmU-25 (-R22-P23-R24-N25-NH$_2$) form multiple polar interactions with the receptor. R24 and N25-NH$_2$ bifurcate at the bottom of the binding pocket and stretch to helices II and IV, respectively (Fig. 2d). N25–NH$_2$ is anchored by hydrogen bonds with E127$^{3.33}$, H185$^{4.64}$, and R288$^{6.55}$, while the positively-charged side chain of R24 forms a salt bridge with E102$^{2.64}$ (Fig. 2d). Mutants such as E127$^{3.33}$A and E102$^{2.64}$A have a complete loss of potency of NmU-25, indicating these residues play important roles in ligand recognition and receptor activation (Fig. 2f, Supplementary Fig. 6, Supplementary Table 2). This is similar to FPR2 and ghrelin receptors, where D106$^{3.33}$ or E124$^{3.33}$ form hydrogen bonds with their corresponding ligands, respectively, and play significant roles in peptide agonist binding and agonist-induced receptor activity[26,31]. Similar recognition mechanisms are also observed in monoamine receptors, which utilize the acidic residues at the 3.32 position to anchor the amine atom of the ligands

and activate the receptor[30]. H185[4.64]F and R288[6.55]A greatly reduced the level of receptor expression, suggesting these residues might play critical roles in mediating the stability of the ligand-binding pockets, which is important for ligand recognition (Supplementary Table 2). Both R24 and N25–NH$_2$ are also involved in hydrophobic interactions with F126[3.32], F284[6.51], and F316[7.42] (Fig. 2d). One of the key residues in receptor activation, F284[6.51], in turn, coincides with the highly conserved residue W281[6.48], triggering the active conformational change of the receptor (Supplementary Fig. 3a, b). Breaking these hydrophobic interactions by replacing F126[3.32], F284[6.51], F316[7.42], or W281[6.48] with alanine strikingly impairs receptor activation (Fig. 2f, Supplementary Fig. 6, Supplementary Table 2). This is most likely caused by the lower affinity of these mutants in comparison with wild-type receptors, however, their lower expression level or efficacy might also contribute to the weaker potency of corresponding ligands. Previous structure-activity relationship studies also revealed that deamination of N25–NH$_2$ or replacement of R24 or N25 with alanine results in inactive peptides, demonstrating that R24–N25–NH$_2$ is indispensable for ligand activity[32]. In addition, P23 is packed with F291[6.58], and R22 is anchored by the salt bridge with E105[2.64]. Breaking the salt bridge between R22 and E105[2.64] compromised the agonist potency of NmU-25 about 70-fold, which is similar to the effect of alanine mutation of F291[6.58], indicating that these interactions also play important roles in NmU-25-induced receptor activation (Fig. 2d, f, Supplementary Fig. 6, Supplementary Table 2). E105[2.64]A also decreased the $E_{max}$ of NmU-25 by 50%, which might be caused by a reduction of the receptor expression level as the receptor was destabilized by the disruption of the hydrogen bonding network (Supplementary Table 2). The above structural evidence suggests that interactions between this hydrophilic portion of the ligand and the receptor play critical roles in mediating the agonism of NmU peptides. Sequence alignment demonstrates that these residues are highly conserved among sapiens NMU1 and NMU2 as well as other species (Supplementary Fig. 4), suggesting a common recognition pattern between the NMUs and their peptide ligands.

Residues F19 and F21 of NmU-25 pack against each other through hydrophobic interactions of the side chain and insert into the hydrophobic pocket formed by helices I, II, VII, and ECL3, while the main chain of L20 is stabilized by forming polar interactions with N109[ECL1] (Fig. 2e). Attenuating these interactions by alanine mutation of residues such as N109[ECL1] and F305[7.31] or mutating A302[7.28] to histidine (residue of NMU1) would lead to the loss of interactions or steric hindrance, which was further supported by the reduced potency of NmU-25 for these mutants (Fig. 2f, Supplementary Figs. 4 and 6, Supplementary Table 2). Residues that interact with this hydrophobic segment share relatively low sequence identity among NMU1 and

NMU2, suggesting that different environments of NMUs exist around this area (Supplementary Fig. 4).

## Recognition between NMU2 and its selective ligands

Two compounds, CPN 116 and CPN 267 were synthesized based on the C terminus of NmU-25 (residues 19–25). These two compounds had different selective preferences (Supplementary Fig. 7a, b)[12,14]. CPN 116 showed high potency for NMU2 (EC$_{50}$ = 6.6 nM) without obvious activation of NMU1, while CPN267 showed high agonistic activity for NMU1 (EC$_{50}$ = 0.25 nM) with 1000-fold selectivity over NMU2[12–14]. In agreement with our conclusion, the chemical structures of CPN 116 and CPN 267 reveal differences in the positions of residues 20 and 21. NMU1 prefers bulky aromatic amino acids in these two positions (W20 and α-methyl-W21 in CPN 267), while NMU2 prefers small alkanes (L20 and L21 in CPN 116) (Supplementary Fig. 7a, b). Close inspection of the NmU-25–NMU2 complex indicates that residues in positions 1.30, 7.28, and 7.32, which interact with L20 and F21 of NmU-25 show less sequence identity in NMU1 and NMU2 (Supplementary Fig. 4). To test whether these residues affect ligand selectivity, we substituted these residues in NMU2 with the respective residues of NMU1 or bulky aromatic residues on the agonistic function of CPN 116 to NMU2. Substitution of H43[1.30] and N306[7.32] with the bulky residue tryptophan compromised the CPN 116-induced signaling effect to vary degrees, indicating the role of these residues in CPN 116 recognition (Supplementary Fig. 7c, Supplementary Table 2). The side chain difference in the ligand binding pockets may underline the receptor selectivity of ligands. Replacement of H43[1.30] or N306[7.32] of NMU2 with glutamate or glutamine, respectively, which are the corresponding residues of NMU1, caused about a 30-fold reduction of the potency of CPN 116 in stimulating NMU2 signaling. A302[7.28]H, which might form a spatial hindrance for peptide binding, also decreased the $E_{max}$ of CPN116 dramatically by about 10-fold (Supplementary Fig. 7c, Supplementary Table 2). On the contrary, these mutants exhibited a better preference toward NMU1 ligands, as N306[7.32]Q showed about 8-fold enhanced potency with an NMU1 selective ligand (CPN 267) compared to the WT receptor (Supplementary Fig. 7d, Supplementary Table 2). In addition, H43[1.30]E also increased the $E_{max}$ of CPN 267 by twofold (Supplementary Fig. 7d, Supplementary Table 2). These results suggest that different residues in 1.30 and 7.32 of NMU1 and NMU2 might play a role in ligand selectivity and provide a structural basis for the future design of receptor-selective drugs.

To explore the binding mode of a non-peptide antagonist, molecular docking of R-PSOP based on the model of NMU2 was also performed. The result suggests that R-PSOP sits into a deeper binding pocket compared with NmU-25 and lies across from helix V to helix II (Fig. 3a). According to the molecular docking, the positively-charged

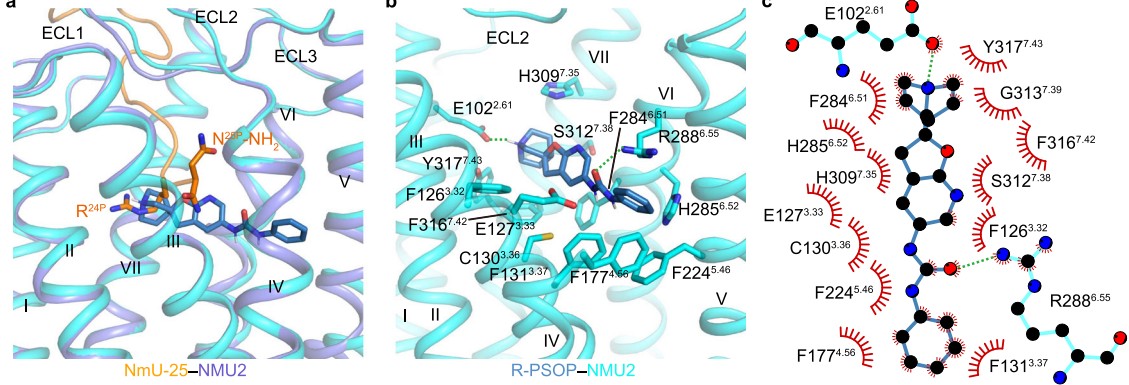

**Fig. 3 | Molecular docking of R-PSOP in NMU2. a** Docking pose of R-PSOP in NMU2. Receptors of the NmU-25–NMU2 complex and R-PSOP–NMU2 docking model are shown as slate and cyan cartoons. Ligands are shown as orange and sky-blue sticks. **b** Close view of the interactions between R-PSOP and NMU2. The residues of the receptor are shown as cyan sticks. Polar hydrogen atoms of R-PSOP are shown as gray sticks. Polar interactions are indicated by green dash lines. **c** Interaction patterns between R-PSOP and the NMU2 represented by LigPlot$^+$ program[64]. Residues of the NMU2 engaged in polar interactions are shown as sticks.

azabicyclo nitrogen of R-PSOP is anchored by polar interactions with E102$^{2.61}$ (Fig. 3b), and the urea group is stabilized through polar interactions with R288$^{6.55}$ (Fig. 3b). In addition, the aromatic ring of azabicyclo, furan, and pyridine are bordered by a hydrophobic environment formed by F126$^{3.32}$, F284$^{6.51}$, and F316$^{7.42}$ (Fig. 3b). The hydrophobic phenyl ring of R-PSOP is inserted into a smaller hydrophobic cavity formed by F131$^{3.37}$, F177$^{4.56}$, and F224$^{5.46}$ of helices III–V, respectively (Fig. 3b). To verify the result of molecular docking and uncover the selectivity mechanism of R-PSOP, we carried out an inositol phosphate (IP) accumulation assay to test the effects of these residues involved in R-PSOP binding. Alanine replacement of E102$^{2.61}$ or F224$^{5.46}$ abolished the antagonist inhibition effect of R-PSOP (Supplementary Fig. 7e, Supplementary Table 3). R288$^{6.55}$A totally eliminated the NmU-25-induced IP accumulation, which may be a consequence of the dramatically reduced expression level (Supplementary Fig. 7e, Supplementary Table 3). These data indicate the important role of polar interactions for ligand recognition. Sequence alignment reveals that residues in the hydrophobic cavity formed by helices III–V are not conserved between NMU1 and NMU2 (Supplementary Fig. 4). L146$^{3.37}$ and C192$^{4.56}$ in NMU1 might weaken the hydrophobic interactions between the phenyl ring of R-PSOP and the receptor. Indeed, the replacement of F131$^{3.37}$ or F177$^{4.56}$ with leucine or cysteine decreased the antagonistic effect of R-PSOP significantly (Supplementary Fig. 7e, Supplementary Table 3), demonstrating the hydrophobic cavity formed by helices III–V might play significant roles in ligand selectivity for the non-peptide antagonist.

## Specific G protein coupling

Since the rapid development of the single-particle cryo-EM, many structures of G protein-coupled GPCRs solved in detergent micelles or lipid bilayers have been obtained[25,33–37]. Superimposition of NMU2 with NTS$_1$ and other G$_i$-bound GPCRs complexes reveals the most notable difference in the complex interface between the receptor and G protein. The αN helix of Gα$_{i1}$ in the NMU2–G$_{i1}$ complex is rotated by about 25° relative to that in the NTS$_1$–G$_i$ complex (Fig. 4a). At the same time, the α5 helix is tilted to a different extent compared with Gα$_{i1}$ in other

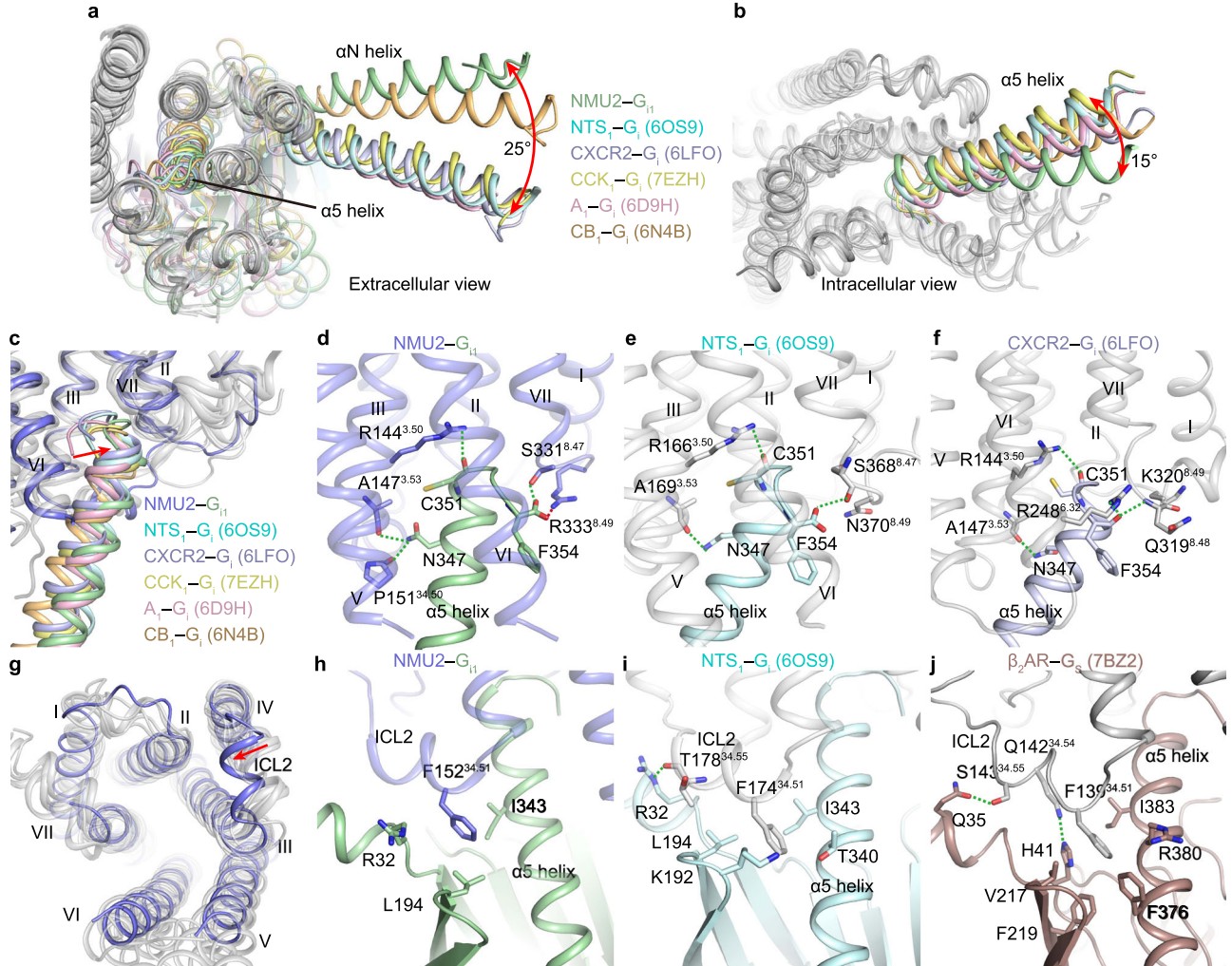

**Fig. 4 | Comparison of G protein coupling of NMU2–G$_{I1}$ complex.**
**a**, **b** Comparison of the G protein binding pose in NMU2–G$_{i1}$, NST$_1$–G$_i$ (6OS9), CXCR2–G$_i$ (6LFO), CCK$_1$–G$_i$ (7EZH), A$_1$–G$_i$ (6D9H) and CB$_1$–G$_i$ (6N4B) complexes. Gα$_i$ in NMU2–G$_{i1}$, NST$_1$–G$_i$, CXCR2–G$_i$, CCK$_1$–G$_i$, A$_1$–G$_i$, and CB$_1$–G$_i$ complexes are shown as cartoon and colored by pale green, pale cyan, blue-white, pale yellow, light pink, and wheat. Gβ and Gγ are not displayed for clarity. Receptors are shown as a gray cartoon. Conformational changes of αN helix (**a**) and α5 helix (**b**) are indicated by red arrows. **c** Comparison of the α5 helix of G$_i$ in the NMU2–G$_{i1}$ and other GPCR–G$_i$ complexes. NMU2 is shown as a slate cartoon, and other receptors are shown as a gray cartoon. Conformational changes of the C-terminus of the α5 helix are indicated by red arrows. **d**–**f** Residues involved in the interactions between the α5 helix and the receptors are shown as sticks. Hydrogen bonds and salt bridges are indicated by green, and red dashed lines. **g** Comparison of the ICL2 in NMU2–G$_{i1}$ and other GPCRs–G$_i$ complexes. The conformational change of ICL2 is indicated by red arrows. **h**–**j** Different binding modes between ICL2 of the receptor and G protein. NMU2 is shown as a slate cartoon, and other receptors are shown as a gray cartoon. G$_s$ of β$_2$AR-G$_s$ complex is shown as a raspberry cartoon.

GPCR–G$_i$ complexes (Fig. 4b). A similar conformational difference in G protein orientation is also observed by the superimposition of other GPCR–G$_i$ complexes with or without scFv16 binding, indicating that the NMU2–G$_{i1}$ complex reflects its real physiological conformation instead of a result of antibody binding (Supplementary Fig. 8)[33,38–42]. These differences might be attributed to variable interactions between receptor and G protein.

Extensive interactions between Gα$_{i1}$ and NMU2 are observed, including residues from αN helix, αN–β1 loop, and α5 helix of G$_{i1}$ and helices II, III, V, VI, VII, and ICL2 of NMU2. For all the interfaces, more contacts are observed between the α5 helix of Gα$_{i1}$ and NMU2 compared to other GPCR–G$_i$ complexes (Fig. 4c–e). Caused by the large conformational changes of Gα$_{i1}$, the C-terminal hook of α5 helix moves closer to helix VII and C terminus of NMU2 (Fig. 4c), leading to the side chain of R333$^{8.49}$, forming a salt bridge with the C-terminal carboxyl of Gα$_{i1}$ (Fig. 4d). Other than this residue, there is no other specific interaction between helix VIII and TMD of the receptor or G protein, resulting in a flexible C terminus of NMU2. Together with another conserved hydrogen bond between the C-terminal carboxyl of Gα$_{i1}$ and S331$^{8.47}$, these two polar interactions help to stabilize the displacement of the C-terminal hook of α5 helix (Fig. 4d). Sequence alignment of all solved class A GPCRs coupled with the G$_{i/o}$ protein reveals that only a small number of receptors have positively-charged residues (arginine or lysine) in 8.49 (Supplementary Fig. 9a). Different from R333$^{8.49}$ in NMU2, the side chain of this residue in other receptors extends to the helical bundle caused by the rotation of helix VIII without interacting with the G protein (Supplementary Fig. 9b–e)[35,36,38,43]. It has been previously reported that the NTS$_1$–G$_i$ complex solved in lipid bilayers constrains the conformation of the receptor and shows the upward movement of helices VII and VIII, which is potentially led by the interactions between the conserved positively charged residue of helix VIII and the phospholipid head groups[33]. The specific interaction between R333$^{8.49}$ of the NMU2 and Gα$_{i1}$ might be affected in lipid bilayers, however, more structural information is needed for further elucidation of the roles of lipids in modulating the conformation of GPCR–G complexes. The association between NMU2 and α5 helix is further stabilized by two additional hydrogen bonds between N347 of Gα$_{i1}$ and A147$^{3.53}$ and P151$^{35.50}$ of NMU2 which are specific in NMU2–G$_{i1}$ complex, and help to stabilize the overall conformation of the α5 helix (Fig. 4d).

Previous works have demonstrated that ICL2 plays a more important role in G$_s$ protein coupling than in G$_i$, and different from the G$_i$ coupling receptors, most G$_s$-coupled receptors have a bulky aromatic residue in 34.51 of ICL2 which is buried by the hydrophobic pocket formed by α5 helix, αN–β1 loop and β2-β3 loop of Gα$_s$[44]. Interestingly, the ICL2 of NMU2 has phenylalanine at the position of 34.51, but the significant translocation of the αN helix leads to the inward movement of ICL2 with reduced hydrophobic interactions between F174$^{34.51}$ of the receptor and G$_i$ (Fig. 4g, h). Similarly, NTS$_1$ also contains phenylalanine at the same position, which mediates weaker hydrophobic interactions in the active state of the NTS$_1$–G$_i$ complex compared with the β$_2$AR–G$_s$ complex (Fig. 4i, j)[25,45]. Thus, the intrinsic difference between G$_i$ and G$_s$ proteins potentially provides a weaker anchoring interaction of ICL2 and allows more diverse receptor–G$_i$ coupling, even though the same aromatic amino acid is present.

Besides coupling to G$_i$, NMU2 also couples to G$_q$ for signaling transduction as well[8]. The key interactions between class A GPCRs and G$_{q/11}$ that play critical roles in maintaining G$_{q/11}$ signaling are mainly mediated by several conserved interactions. It has been reported that L358$^{H5.25}$, the conserved C terminus residue at the C-terminal "wavy hook" of G$_{q/11}$, forms hydrophobic interactions with V/A$^{6.33}$ and L$^{6.37}$ of their corresponding receptor. Two non-conserved residues of G$_q$, Y356$^{H5.23}$, and N357$^{H5.24}$, form polar interactions with D$^{3.49}$, R$^{3.50}$, N/S$^{8.47}$,

or N/R$^{8.49}$ of the receptor. In addition, a highly conserved hydrophobic residue (I, L, or F) and a hydrophilic residue in ICL2$^{45.51}$ and ICL2$^{45.54}$ also mediate interactions with G$_{q/11}$[46–53]. All the above residues appear to be conserved in NMU2, indicating that NMU2 bind to G$_q$ in a similar manner to other GPCRs. Collectively, the specific interactions between the NMU2 and G$_{i1}$ intrigue the NMU2 to engage G protein with the specific conformational state, revealing a diversified GPCR–G$_i$ coupling mode. This will provide a more structural basis for understanding the complexity of GPCRs and G$_i$ protein coupling.

## Discussion

Since the first discovery of NmU from the porcine spinal cord, NmU peptides have been subsequently identified with multiple physiological roles over the past three decades[5,6,11]. The highly homologous and asparagine amidated C terminus of NmU peptides are vital for receptor binding and functions, but the molecular mechanisms are still unknown. Here we determine the structure of the NmU-25–NMU2–G$_{i1}$ complex and provide a detailed binding mode of NmU-25 with NMU2, depicting a comprehensive mechanism of which NmU peptides are recognized by their cognate receptors. As for the high sequence identity between NMU1 and NMU2, most residues residing in the orthosteric binding pocket are conserved. However, combined with mutagenesis data, we demonstrate that an unconserved pocket exists in helices I and VII, which affects the potency of selective agonists for NMU2. Moreover, swapping residues of this pocket in NMU2 for those of NMU1 enhanced the NMU2 response to the NMU1 selective agonist. This evidence demonstrates the important role of these residues in ligand selectivity for NMUs. In addition, a receptor-selective antagonist binding mode was illustrated by molecular docking and further verified by functional assays. These results demonstrate the diversity of ligand binding modes as well as a selectivity mechanism of NMU2.

A number of GPCR–G protein complexes have been reported due to the development of cryo-EM technology. These complexes revealed activation of the receptor, diverse G protein binding modes, G protein coupling selection as well as different conformational states. Structural comparison of the NMU2–G$_{i1}$ complex with other solved GPCR–G$_{i1}$ complexes suggests a conserved activation process of NMUs in combination with a non-classical G protein binding mode of NMU2. A larger rotation of Gα$_{i1}$ in the NMU2–G$_{i1}$ complex reveals a specific G protein binding mode. Furthermore, 3D variance analysis was performed on the dataset of the NMU2–G$_{i1}$ complex using cryoSPARC to assess dynamics within the NMU2–G$_{i1}$ complex (Supplementary Movie 1). 3D variance analysis showed rocking motions on an axis parallel to the membrane between NMU2 and G$_{i1}$, which was similar to a previous study on the CB$_1$–G$_i$–scFv16 complex[54]. It is interesting that NMU2 and CB$_1$ showed a different rotation of the αN helix in comparison with other GPCR–G$_i$ complexes. These results suggested the intrinsic flexibility and heterogeneity of GPCRs for transient protein interactions.

Altogether, these findings enhance our understanding of the molecular mechanisms of ligand recognition, selectivity, and activation of NMU2, providing a reliable structural framework for rational drug design to target this receptor.

## Methods
### Construct cloning and protein expression
The gene of human NMU2 was cloned into a modified pFastBac1 vector containing a HA signal peptide followed by a Flag tag at the N terminus. The C-terminal residues Q356-T415 of NMU2 were replaced with a PreScission protease (PPase) site along with a twin-strep tag to improve the stability of the NMU2–G$_{i1}$ complex. The gene of dominant-negative human Gα$_{i1}$ (DNGα$_{i1}$) containing five mutations (S47C, G202T, G203A, E245A and A326S) was cloned into the pFastBac1

vector[55]. The genes of human $G\beta_1$ and $G\gamma_2$ with a 6 × histidine (His) tag at the N terminus of $G\beta_1$ were cloned into the pFastBac Dual vector (Invitrogen). High-titer recombinant baculovirus was obtained using the bac-to-bac baculovirus expression system. High-Five insect cells at a cell density of $1.5 \times 10^6$ cells/ml were infected with modified NMU2, $DNG\alpha_{i1}$, and $G\beta_1\gamma_2$ at an MOI (multiplicity of infection) ratio of 1:1:1. The cell pellets were collected by centrifugation after 48-h post transfection and stored at −80°C until use.

## Purification of NmU-25–NMU2–$G_{i1}$ complex
Cells were disrupted by a homogenizer in lysis buffer containing 25 mM HEPES, pH7.5, 150 mM NaCl, 10 mM $MgCl_2$, 10% (v/v) glycerol, and EDTA-free complete protease inhibitor cocktail. To assemble the complex in the membrane, 50 µM NmU-25, 50 µM TCEP, and 100 mU/ml apyrase (NEB) was added to the membrane and incubated at 20 °C for 1 h. The complex was extracted by adding 0.5 % (w/v) lauryl maltoseneopentyl glycol (LMNG, Anatrace), 0.05 % (w/v) CHS at 4 °C, and incubated for 3 h. The supernatant was collected by centrifugation and then incubated with Strep-Tactin XT Superflow resin (IBA Lifesciences) overnight at 4 °C. The resin was washed with a buffer containing 25 mM HEPES, pH 7.5, 150 mM NaCl, 0.01% (w/v) LMNG, 0.001% (w/v) CHS, and 25 µM NmU-25. The complex or receptor was eluted with a buffer containing 150 mM Tris, pH 8.0, 150 mM NaCl, 0.01% (w/v) LMNG, 0.001% (w/v) CHS, 50 µM NmU-25, and 50 mM biotin (Sigma-Aldrich) and then incubated with Ni-NTA Superflow resin supplemented with 5 mM imidazole at 4 °C for 2 h. The resin was washed again with a buffer containing 25 mM HEPES, pH 7.5, 150 mM NaCl, 0.01% (w/v) LMNG, 0.001% (w/v) CHS, 10 mM imidazole, and 25 µM NmU-25. The complex was eluted with a buffer containing 25 mM HEPES, pH 7.5, 150 mM NaCl, 0.002% (w/v) LMNG, 0.0002% (w/v) CHS, 300 mM imidazole and 50 µM NmU-25. Comparison of the NmU-25–NMU2–$G_{i/q/11}$ complex formation in vitro using analytical size-exclusion chromatography showed the most promising complex behavior, and thus, $G_{i1}$ was chosen for structural studies. The NmU-25–NMU2–$G_{i1}$ complex was further purified by size-exclusion chromatography using a Superdex 200 Increase 10/300 column (GE Healthcare). Peak fractions of the complex were collected and concentrated to 3–4 mg/ml with a 100-kDa molecular weight cut-off concentrator (Millipore) for cryo-EM studies.

## Cryo-EM data collection
The purified NmU-25–NMU2–$G_{i1}$ complex was diluted to 2 mg/ml and applied to glow-discharged holey carbon grids (CryoMatrix M024-Au300-R12/13). The grids were blotted at 4 °C, 100% humidity with the force of 0 and blot time of 1.0 s and plunge-frozen in liquid ethane using Vitrobot Mark IV (ThermoFisher Scientific) and stored in liquid nitrogen until use. Images were obtained by a Titan Krios G3 electron microscope (FEI) of 300 kV with K3 Summit direct electron detector (Gatan) using a pixel size of 1.045 Å. A total of 8,788 movie stacks were recorded with defocus ranging from −0.8 to −1.5 µm, exposing them to a dose rate of 1.875 electrons/Å²/frame. Each movie stack contains 40 frames for a total dose of 70 electrons/Å². SerialEM[56] was applied to automated single-particle data acquisition.

## Cryo-EM data processing
Collected movies were subjected to beam-induced motion correction and contrast transfer function determination by MotionCor2[57] and Gctf v1.18[58]. Total of 3,088,911 particles of NmU-25–NMU2–$G_{i1}$ complex were auto-picked using RELION 3.1 and then subjected to three rounds of reference-free 2D classification to discard false-positive particles[59]. An ab initio model generated by RELION 3.1 was used as an initial reference model for 3D classification. The best class with 912,031 particles of NmU-25–NMU2–$G_{i1}$ complex was selected and subjected to 3D auto-refinement in RELION3.1 in succession. The final maps were improved by Bayesian polishing, resulting in a 3.3 Å

density map based on the gold-standard Fourier shell correlation using the 0.143 criteria. The local resolution for the map was generated by ResMap[60].

## Model building
The receptor of the NmU-25–NMU2–$G_{i1}$ complex was built using $NTS_1$ (PDB ID: 6OS9) as the template by SWISS-MODEL. Subunits of $G_i$ were built using the components of glucagon–GCGR–$G_i$ complex (PDB ID: 6LML). Models were fitted into electron density maps using UCSF Chimera[61]. Subsequently, models were merged and rebuilt using COOT, refined by PHENIX, respectively. The final model was validated by MolProbity[62]. Structure figures in this paper were prepared by Pymol (https://pymol.org/2/) and UCSF Chimera.

## Molecular docking of R-PSOP
The 3D structure of R-PSOP used for docking was downloaded from PubChem (ID: 73755058) as sdf file. The protonated state of R-PSOP was predicted by the Epik module with a pH of $7.5 \pm 2$ and optimized using the Ligprep tool of the Schrödinger suite. The pKa values of ionizable groups in the receptor of the NmU-25–NMU2–$G_{i1}$ complex were predicted at pH 7.5. The receptor was prepared using the Protein Preparation Wizard implemented in the Schrödinger suite to add the missing side chains and hydrogen atoms. The overall structures of the receptor were refined using an OPLS3 forced field based on the heavy atoms restraint. Docking of R-PSOP to NMU2 was performed with the Ligand Docking, and the rotatable groups were selected among all allowed groups by Glide that potentially interacted with the ligand. The top energy minimized docking pose of protonated R-PSOP was selected according to GildeScore for further Induced Fit Docking[63]. Induced Fit Docking was done at default settings, except the extra-precision mode was selected in the "Glide Redocking." In the "Prime Refinement," all residues at the distance of 5 Å from the ligand pose were refined, and no other residues were specified for refinement. The docking grid was centered on the centroid of the ligand from the top energy-minimized docking pose. The docking pose of R-PSOP was selected from high-score conformations with reasonable binding mode for further structural analysis.

## TRUPATH assay
The plasmids of $G\alpha_{i1}$, $G\beta_3$, and $G\gamma_9$ used for the TRUPATH assay were purchased from Addgene (TRUPATH Kit, #1000000163). NMU2 constructs of WT and mutants were constructed into a PTT5 vector (Invitrogen) with an N-terminal HA signal peptide followed by a Flag tag. HEK 293F cells were transiently transfected with WT or mutants of NMU2, $G\alpha_{i1}$, $G\beta_3$, and $G\gamma_9$ at an MOI ratio of 4:1:1:1 with a total of 3500 ng plasmid DNA using a transfection reagent (PEI MAX 2000, Polysciences) and cultivated at 37 °C with 5 % $CO_2$. Cells were collected after 48-h transfection. The expression level of the receptor was measured by incubating 10 µl cells with 15 µl TBS buffer supplemented with 4% bovine serum albumin, 20% (v/v) viability staining solution7-AAD (Invitrogen, Cat#00-6993-50), and ANTI-FLAG M2-FITC antibody (Sigma, F4049; 1:100 diluted by TBS) at 4 °C for 20 min. After incubation, 175 µl TBS buffer was added, and the fluorescence signal on the cell surface was detected by an FCM (flow cytometry) reader (Millipore). The cells were diluted with 1× HBSS (Hank's balanced salt solution) balanced buffer supplemented with 25 mM HEPES, pH 7.4, and seeded onto the 96-well opaque cell culture plates (Beyotime) at a density of 18,000 cells per well by 60 µl. Freshly prepared 50 µM coelenterazine 400a (Nanolight Technologies) was added into the well by 10 µl. After a 5 min equilibration time in the darkness, plates were then read in a Synerg™ H1 microplate reader (BioTek) with excitation at 395 nm and emission at 410 and 515 nm by serially three times. Then, 30 µl of 3.3× agonists were added into the wells with a gradient final concentration from $10^{-4}$–$10^{-13}$ M. Plates were read immediately again by serially five times.

Data were processed and analyzed following the method in the literature by using GraphPad Prism 8.0.

## IP accumulation assay

NMU2 constructs of WT and mutants were constructed into PTT5 vector (Invitrogen), as mentioned above. HEK 293F cells were transiently transfected with WT or mutants with total of 2000 ng plasmid DNA using transfection reagent (PEI MAX 2000, Polysciences) and cultivated at 37 °C with 5% $CO_2$. Cells were collected after 48-h transfection. IP accumulation was tested by IP-One Gq assay kit (Cisbio Bioassays, 62IPAPEJ) following the instruction manual. In general, cells were resuspended with 1× HBSS buffer containing 20 mM LiCl (stimulation buffer) and seeded into the white 384-well microplates (Proxi Plate™-384 Plus, PerkinElmer) with a density of 20,000 per well. Cells were preincubated with $10^{-5}$ M R-PSOP (provided by Boehringer Ingelheim company, Germany) and gradient concentration from $10^{-4}$ to $10^{-11}$ M of NMU-25 or gradient concentration $10^{-4}$ to $10^{-11}$ M of NMU-25 alone in stimulation buffer at 37 °C for 90 min. The cryptate-labeled anti-IP1 monoclonal antibody and d2-labeled IP1 were diluted with lysis & detection buffer (1:20) and added to each well by 3 μl, respectively. The plates were incubated at room temperature for 60 minutes and then read in a Synerg™ H1 microplate reader (BioTek) with excitation at 330 nm and emission at 620 and 665 nm. The IP1 production was calculated by a standard dose–response curve and analyzed by GraphPad Prism 8.0.

## Reporting summary

Further information on research design is available in the Nature Portfolio Reporting Summary linked to this article.

## Data availability

Atomic coordinate and the cryo-EM density map of NmU-25–NMU2–$G_{i1}$ complex have been deposited in the RCSB Protein Data Bank (PDB) under accession code 7XK8, and Electron Microscopy Data Bank (EMDB) under accession code EMD-33247. Source data are provided in this paper. All relevant data are available from the corresponding authors upon request. Source data are provided in this paper.

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

## Acknowledgements

Cryo-EM data collection was performed at the EM facility of the Shanghai Institute of Materia Medica (SIMM), Chinese Academy of Sciences. We thank Q. Wang from SIMM for the data collection. This work was supported by the National Key R&D Program of China 2018YFA0507000 (Q.Z. and B.W.), CAS Strategic Priority Research Program XDB37030100 (Q.Z. and B.W.), and National Science Foundation of China grants 31825010 and 82121005 (B.W.).

## Author contributions

W.Z. prepared the protein sample for cryo-EM studies, designed the functional assays, performed the IP1 accumulation assay, and prepared the manuscript. W.Z. helped with protein sample preparation and performed the TRUPATH assay. M.W. performed the docking studies. M.L. helped with the functional and IP1 accumulation assay. S.C. performed the TRUPATH assay and helped with paper preparation. T.T. helped with protein sample preparation. G.S., H.W., and A.B. helped with the preparation of R-PSOP. C.Y. and X.C. expressed the proteins. S.H. prepared cryo-EM samples, collected cryo-EM data, performed cryo-EM data processing, solved cryo-EM structure, and wrote the first draft. Q.Z. and B.W. initiated the project, planned and analyzed experiments, supervise the research, and wrote the paper with input from all co-authors.

## Competing interests

The authors declare no competing interests.
