## [Peer Review File · Nature Communications]

Ligand recognition and activation of neuromedin U receptor 2REVIEWER COMMENTS

Reviewer #1 (Remarks to the Author):

The manuscript by Zhao et al. reports a cryo-EM structure of NMUR2 bound to endogenous agonist NMU25 and Gi1. Moreover, the Authors used molecular docking to explore the binding mode of non-peptide antagonist of this receptor.

The manuscript can be published after considering the following suggestions:

1. Regarding protein preparation for molecular docking in the Protein Preparation Wizard, did the Authors predict the pKa values of ionizable groups in protein with PROPKA of the Schrödinger suite?
2. The non-peptide antagonist R-PSOP has a protonable nitrogen atom. The protonation state should be assigned with the Epik module of the Schrödinger suite.
3. Accordingly, in Fig. 3 polar hydrogen atoms of the ligand should be shown.
4. Regarding molecular docking with Glide, did the authors allow for rotation of some side chains at the stage of grid generation? This should be done and rotatable groups should be specified.
5. Regarding induced-fit docking, was it done at default settings? Which residues were refined? At the distance of 5 Å from the ligand pose or others?
6. As for the selection of the ligand pose: relying only on docking score is elusive. The final pose should be selected based on careful visual inspection.
7. The discussion section is rather brief and should be extended.
8. Minor issues: please correct some grammar errors, typos, add missing symbols and issues like doesn't (does not).

Reviewer #2 (Remarks to the Author):

In their manuscript by Zhao et al, the authors report the structure of the neuromedin 2 receptor bound as a complex to a heterotrimeric G protein and agonist NMU25. The cryoEM structure is of relatively high resolution and provides useful structural information regarding the binding site for the natural peptide agonist, NMU25, as well as inferences drawn from the structure on small molecule ligands. Docking of the selective NMUR2 antagonist R-PSOP on to the new NMUR2 structure provides a structural rationale for the specificity and thus has implications for future drug development of novel NMUR2 modulators, as well as development of NMUR1 selective ligands. The authors supply some mutagenesis and functional data to validate the structural models for both the peptide agonists as well as small molecule antagonist. The structure reveals subtle changes in the conformation of the Galphai-subunit of the heterotrimer, revealing a significant rotation of the Galphai N-terminus and small

differences in the angle of attack of the C-terminal α 5-helix, in comparison with other Gi-coupled receptor complexes.

This is a solid study on a novel GPCR-Gi complex and offers insight into the structure-activity relationships of natural and synthetic modulators. However, the report does not really offer novel insight into how ligand binding leads to G protein activation or suggest how the slightly different structures represent a unique mechanism to promote GDP release from Galpha. Nor do the authors really address ligand efficacy: why in fact the peptides are agonists and why the small molecule R-PSOP is an antagonist? It is a very nice but descriptive study of another cryoEM structure of G protein complex with a GPCR superfamily member.

Comments:

The authors reveal structural changes in the G protein alpha subunit's N- and C-terminus. In their structure the N-terminus appears rotated in comparison to Galpha complexes bound to other GPCRs. In addition, the authors observe a subtle difference in the angle of approach of the alpha5 helix (C-term) of Galpha, compared to other G protein-receptor complexes. Some care should be taken before making these comparisons. Several of the referenced structures took advantage of an antibody that stabilized the interaction between the N-term of Galpha and the Gbeta subunit. This antibody fragment was critical to these studies to ensure tight association of the trimer while still allowing G protein coupling. The authors did not use this reagent to resolve their structure. Thus, it is not clear whether the NMUR2-Gi complex structure is closer to the native conformation or whether the non-stabilized complexes are more anisotropic. It would be helpful to know whether the resolution around these regions is different compared to other antibody-stabilized complexes.

A related issue concerns whether there is something special about the NMUR2 and CB1 that allows the Gi N-termini to adopt this rotated conformation? It is pretty interesting that these two receptor-Gi complexes appear as outliers.

It is curious why the authors used the GCGR-Gi complex to build their model but did not include the GCGR-Gi structure in their superposition with other Gi-coupled receptors. How do these N and C-termini positions differ in the GCGR-Gi complex relative to the NMUR2-Gi structure?

The authors appropriately included cholesterol hemisuccinate in their extraction, purification and imaging buffers. The protein concentration steps are appropriate and utilize cut-offs that should not concentrate empty micelles. The question is whether the receptor-G complex structures that the authors are comparing also include similar steps where cholesterol in particular is included at similar concentrations. Cholesterol and CHS are known to bind to regions surrounding TM7 and helix 8. Since

these regions are critical for G protein coupling it is important to compare receptor complex stabilized in similar experimental conditions. It should also be noted that many of these differences might disappear if the receptor-G protein complexes were resolved in a lipid bilayer and not in detergent micelles.

In the description of the critical aspects for NMU25 C-terminus in efficacy of peptide ligand, the authors comment about the importance on the Asn25 amidation. It's important that the authors draw similarity to other monoamine receptors and their engagement of Asp3.33 or Glu3.33 for both binding affinity as well as agonist efficacy.

The authors note that agonist binding does not directly engage TM5, which is interesting. Many monoamine receptor engage TM5, as well as other TMs, that result in collapse of the core of the ligand binding, ultimately resulting in rotation and translation of TM6. It might be useful to include a superposition of TM core of NMUR2 with other active state GPCRs.

Reviewer #3 (Remarks to the Author):

This paper reports a new cryo-EM structure of the NMUR2 complexed with peptide agonist and Gi protein. The structure appears to be reasonable resolution and the authors state all modelling is unambiguous. This appears to be the case given the resolution of the map and the map to model figures shown in supplementary Fig 2, but without the maps and pDBs, it is difficult to verify this. Nonetheless the structural work appears to be well done. Mutagenesis data is also provided to support the interactions of NMU-25 with the receptor that were identified in the structure, and the authors provide a structural hypothesis for ligand selectivity to the NMUR2 over the closely related NMUR1.

While the data are certainly interesting, to be informative for the design of selective agonist for NMURs additional data is required to draw definitive conclusions on the structural basis of selectivity. While there is mutagenesis data in this manuscript on NMUR2 (introducing NMUR1 residues into NMUR2) that provide some support for key residues for selectivity, further evidence is warranted. For example, there is no structural, modelling or functional data provided for NMUR1 in the manuscript. Ideally structural data with selective ligands bound to each receptor, as well as non-selective ligands would be provided. However, in the absence of structure, minimally, if the identified residues are key to selectivity, it would be important to perform the reciprocal mutagenesis in NMUR1 and assess the selective NMUR2 agonist at these mutations (ie do they gain affinity over the WT NMUR1). Moreover, the selective NMUR1 agonist that is shown in this paper is not actually assessed at the introduced mutations in NMUR2 to assess if the potency of this agonist is increased when the NMUR1 residues are introduced.

Interestingly the authors identify a conformation of Gi2 that differs to the conformation observed in other Gi bound structures (relative to the receptor), including the closely related NTS1R. However, two conformations of Gi were observed in the NTSR1 structure. How does the conformation identified here compare to the second conformation observed for Gi bound to the NTSR1? Is there any evidence of other conformations of Gi in the NMUR2 dataset (ie only 900 000 particles of 3 million that were selected were used in the final reconstruction). Numerous studies have used 3D variance analysis to assess dynamics within complexes, and these reveal common rocking motions of the receptor relative to G protein that could lead to differences in consensus density maps depending on how the particles are selected. 3D variance analysis should be performed on this dataset to observe if the more common Gi conformation is present among the conformational space sampled at the time of vitrification.

It is also interesting that the authors did not see density in the cryo-EM map for helix 8. Why do the authors think this is the case? Do they think that the receptor lacks helix 8, or that it becomes unstructured in the presence of Gi? It would be interesting to assess this in the 3D multivariate analysis to see if there is any evidence within the particle stack of the presence of H8. It is also important to note that the structure is in detergent and therefore H8, the α N and β g subunits of the G protein might be influenced differently in detergent vs lipid, ie the different Gi conformation could be an artefact of detergent extraction.

Can the authors clarify why they selected Gi1? This receptor couples to other Gi subunits as well as Gq. Given all the functional data was performed using an IP1 assay (downstream of Gq), why was Gq not used in the structural studies?

Mutagenesis was used to validate the interactions between ligand and receptor. All changes in potency of the mutant receptors have been interpreted as reduced interaction of the ligands with the receptor. However, in a large number of the mutants, receptor expression was significantly lower than WT. Given a functional assay (which is influenced by receptor expression levels) was used to validate the mutants, how can the authors distinguish whether the effects on potency are related to the lower expression or reduced interactions of the ligands with the receptor?

Figure 2 – panels e and f are labelled incorrectly.

Line 168-169 – 100-fold selectivity for NMUR2 – should this read over NMUR2????

Supp Fig 6c – should this be [CCPN116] on the x axis, not [NMU-25]?

Line 204 references a TRUPATH assay, yet none of the figures referenced show TRUPATH data.

There is no method reported for the flow cytometry assay used to measure receptor expression and there are a large number of grammatical errors throughout the manuscript that will need to be addressed.

Reviewer #4 (Remarks to the Author):

This manuscript presents some very nice and important structural information relating to NMU2, its interaction with its ligands (endogenous NmU25 and synthetic small molecules) and binding to a G-protein alpha subunit. The information is very pertinent, particularly as the authors point out, in a drug discovery context. There is little doubt that this receptor presents an interesting therapeutic target for the treatment of conditions such as obesity and that the generation of suitable (possibly small molecule) ligands has been somewhat hampered by a lack of structural information and particularly a lack of understanding about ligand-receptor interaction and receptor activation. The manuscript presents important and interesting information about the recognition and binding of NmU-25 to NMU2, the activation mechanisms of NMU2 particularly by an endogenous ligand (NmU-25), along with some key information relating to G-protein coupling and receptor selectivity shown by synthetic ligands. Thus, this manuscript addresses important questions relating to NMU2 and provides more general insight into GPCR activation and functioning generally.

There are a number of typographical and grammatical errors that require attention. Furthermore, some aspects require a little work to improve accuracy and to help the reader a little more. This could include more helpful statements about the roles of NmU ie. where possible, be explicit in the roles rather than simply stating regulates, elicits etc. The figures are very nice and present the information clearly. I have one or two queries about the colour references in the figure legends, which themselves would also benefit from a little editing.

It would be prudent to use the IUPHAR recommended abbreviations for the receptors and ligands: <https://www.guidetopharmacology.org/GRAC/FamilyDisplayForward?familyId=42>

Ensure abbreviations defined at first use eg. NTSR1, β 2AR.

Accession codes are missing.

Some care needs to be taken with the terminology around changes in ligand potency associated with mutagenesis. For example, breaking the salt-bridge between R22 of the ligand and E105 by mutating to

alanine, is said to decrease the EC50 about 50-fold. This mutation increases the EC50 (37 nM in WT to 1854 nM in E105A). The pEC50 and agonist potency are reduced. Similarly, the mutations N109A, F305A and A302H increase rather than decrease the EC50. Additionally, when discussing CPN116 and CPN267, the authors switch between potency and affinity. As affinity has not been measured, it is not possible to directly attribute changes in potency to changes in affinity; the data could certainly suggest or support this (and should be phrased appropriately). Replacement of H43 or N306 do not decrease the EC50; they increase the EC50 and lower potency.

NMU2 is clearly able to couple to Gi. There is functional experimental evidence to support this and it is not uncommon for Gs- and Gq-coupled receptors to show such promiscuity. However, given that NMU2 is preferentially a Gq-coupled receptor, some comment about the choice of G-protein in the present study would be warranted, along with at least some discussion of the relevance and any potential consequence of Gq coupling in relation to the present findings. For example, given the C-terminus of the G-protein is widely considered to be responsible for driving recognition by the receptor, are there any consequences of using Gi rather than Gq to determine the binding and activation mechanisms? Are different conformations required for Gi and Gq coupling?

Although the determined structures and the bulk of the pharmacology (using the TRUPATH assay) have focused on Gi-coupling, the data around the antagonism of R-PSOP addresses IP accumulation and therefore Gq-coupling. Are there any issues around switching between different cellular outcomes with different transduction pathways?

One of the difficulties when assessing the impact of mutation on the coupling/function of receptors is interpreting the potential confounding effects of changes in cell surface expression. It was reassuring to note that the authors have assessed expression but differences in expression were not addressed as part of the consideration of the effect of mutation on function. For example, in the data from the TRUPATH assay, there was a fairly notable positive relationship between the expression of the mutant receptor (compared to wild-type) and the Emax values. Is there any possibility that one or more of the mutants impacted on, for example, receptor stability and/or receptor trafficking? Over what range of expression levels are wild-type responses consistent (ie. similar in EC50 and Emax) and does this include the range covered by the expression of the mutant receptors? Do these points merit discussion? As specific examples, it is suggested that mutations of either N109, A302 or F305 decrease agonist potency. There is a presumption that this is due to reductions in ligand affinity but this has not been determined. Changes in EC50 may result from changes in affinity, intrinsic efficacy and receptor expression levels.

For some of the mutants, there were significant effects on agonist potency in the TRUPATH assay. Again, is it possible that any of the shift in EC50 values were a consequence of alterations in expression? Further, where mutation reduced agonist potency without impacting receptor expression (e.g.H43E and N100A), would the data suggest that this may be a consequence of reduced ligand affinity or a reduction in ligand efficacy? Perhaps a little more discussion and clarity would add to the manuscript.

In supplementary figures 5 and 6, it perhaps should be stated that the lowest concentration of ligand used was either zero or -11 (\log_{10} M). I appreciate the difficulties of expressing zero on this scale.

Would it be worth commenting a little further on why the specific receptors have been selected for more detailed comparisons (NTSR, OX2R) and the significance of these?

Whilst the manuscript has focused on NmU-25 as the endogenous ligand, highlighting the consistency of the C-terminus of this and other peptide ligands, do the data allow any discussion of potential roles for the N terminus and in particular differences here between the endogenous ligands (ie. NmU and NmS)?

Generally, the Discussion is somewhat limited and would benefit from a wider consideration of the findings.

Might it be useful for Supplementary Figure 4 to contain an indication of the location of the TMDs, ECLs and ICLs?

Point-by-Point response to reviewer comments

We thank reviewers for the constructive comments they provided. We have answered all questions in detail, and changes made in the manuscript are described below.

Reviewer #1 (Remarks to the Author):

The manuscript by Zhao et al. reports a cryo-EM structure of NMUR2 bound to endogenous agonist NMU25 and Gi1. Moreover, the Authors used molecular docking to explore the binding mode of non-peptide antagonist of this receptor.

The manuscript can be published after considering the following suggestions:

1. Regarding protein preparation for molecular docking in the Protein Preparation Wizard, did the Authors predict the pKa values of ionizable groups in protein with PROPKA of the Schrödinger suite?

-- We thank the reviewer for this comment. We used the PROPKA module in the Protein Preparation Wizard to predict the pKa values of ionizable groups in protein with pH of 7.5 as input. Followed the reviewer's suggestion, the statement "The pKa values of ionizable groups in the receptor of the NmU-25–NMU2–Gi1 complex were predicted at pH 7.5." has been added to the method of "Molecular docking of R-PSOP" in line 410-411.

2. The non-peptide antagonist R-PSOP has a protonable nitrogen atom. The protonation state should be assigned with the Epik module of the Schrödinger suite.

--We thank the reviewer for this reminder. Yes, the protonation state of the nitrogen atom was predicted by the Epik module at pH of 7.5 ± 2 . Pre-set R-PSOP with the protonated and unprotonated nitrogen atom from LigPre were both subjected to Glide for docking. One R-PSOP binding pose with the protonated nitrogen atom was chosen for further induced-fit docking. For clarity, the statement "The protonated state of R-PSOP was predicted by the Epik module with a pH of 7.5 ± 2 " has been added to the method of "Molecular docking of R-PSOP" in line 408-409.

3. Accordingly, in Fig. 3 polar hydrogen atoms of the ligand should be shown.

--We followed the reviewer's suggestion and corrected Fig. 3a, b by showing the polar hydrogen atoms of the ligand.

4. Regarding molecular docking with Glide, did the authors allow for rotation of some side chains at the stage of grid generation? This should be done and rotatable groups should be specified.

--We appreciate the reviewer for this comment. We originally intended to detect possible binding poses roughly by Glide then further refined the possible interactions in the induced-fit docking. For more accurate docking, we followed the reviewer's advice

and regenerated the grid by allowing rotatable groups. Very similar docking results by allowing rotatable group were analyzed and possible pose was selected by visual inspection and then sent to induced-fit docking for double check. There was a similar result compared with previous docking result (see above figure). For clarity, the statement “and the rotatable groups were selected among all allowed groups by Glide that potentially interacted with the ligand” has been added to the method of “Molecular docking of R-PSOP” in line 415-416. Fig. 3 has been updated by the new docking result.

5. Regarding induced-fit docking, was it done at default settings? Which residues were refined? At the distance of 5 Å from the ligand pose or others?

-- Induced-fit docking was done at default settings except that the extra-precision mode was selected in the “Glide Redocking”. In the “Prime Refinement”, all residues at the distance closer than 5 Å from the ligand poses were refined and no additional residues were specified for refinement. These descriptions above have been added to the method of “Molecular docking of R-PSOP” for clarity. “Induced Fit Docking was done at default settings except the extra-precision mode was selected in the “Glide Redocking”. In the “Prime Refinement”, all residues within the distance of 5 Å from the ligand were refined and no other residues were specified for refinement.” in line 418-421.

6. As for the selection of the ligand pose: relying only on docking score is elusive. The final pose should be selected based on careful visual inspection.

-- We thank the reviewer for this reminder. Two types of distinct binding poses were observed after docking. Both docking score and visual inspection were considered. The final pose shown in the Fig. 3 was selected with the reasonable binding mode: the hydrophilic part formed polar interactions with the receptor while the hydrophobic part protruded into the hydrophobic pocket formed by helices IV and V, which also had the highest docking score.

7. The discussion section is rather brief and should be extended.

--We followed the reviewer’s suggestion and extended the discussion section at page 11-12.

8. Minor issues: please correct some grammar errors, typos, add missing symbols and issues like doesn’t (does not).

-- We thank the reviewer for this comment. We have corrected these errors in the manuscript carefully.

Reviewer #2 (Remarks to the Author):

1. The report does not really offer novel insight into how ligand binding leads to G protein activation or suggest how the slightly different structures represent a unique mechanism to promote GDP release from Galpha. Nor do the authors really address ligand efficacy: why in fact the peptides are agonists and why the small molecule R-PSOP is an antagonist? It is a very nice but descriptive study of another cryoEM

structure of G protein complex with a GPCRS superfamily member.

--We thank the reviewer for the comment. However, we disagree with the reviewer's opinion about novel finding of the manuscript. After analyzing the cryo-EM structure of NMU2, both the ligand activation mechanism as well as how ligand selectivity was achieved between NMU1 and NMU2 was discovered. The key residues that contributing to subtype selectivity was first identified, and together with the definition of peptide binding pocket, it will provide the structural basis for designation of novel and high selective drugs. However, as for the receptor activation, it adopts a more common activation process with the conformational changes of PIF and NPXXY motifs. In addition, to explain the antagonism of small molecule R-PSOP, it will be more accurate to obtain the structure of NMU2-R-PSOP. The docking of antagonist in might only provide an antagonist bound receptor in active state, and thus the analysis of antagonism might be misled by the difference of receptor activation state. So we hope that the reviewer to agree with not to over emphasize this part of the finding to avoid a redundant description about receptor activation as previous studies and also avoid to distraction of the readers.

In the meanwhile, we agree with the reviewer that we should strengthen our findings about NMU2 complexes. So description about the different activation mode of different motifs used was added to the manuscript as stated in response 6 and 7 to Reviewer #2. Also, the selectivity of NMU2 was also strengthened by new data of NMU1 and NMU2 selective ligands against different mutants of this receptor. Please also see response to comment 1 of Reviewer # 3.

2. The authors reveal structural changes in the G protein alpha subunit's N- and C-terminus. In their structure the N-terminus appears rotated in comparison to Galphai complexes bound to other GPCRs. In addition, the authors observe a subtle difference in the angle of approach of the alpha5 helix (C-term) of Galphai, compared to other G protein-receptor complexes. Some care should be making before making these comparisons. Several of the referenced structures took advantage of an antibody that stabilized the interaction between the N-term of Galphai and the Gbeta subunit. This antibody fragment was critical to these studies to ensure tight association of the trimer while still allowing G protein coupling. The authors did not use this reagent to resolve their structure. Thus, it is not clear whether the NMUR2-Gi complex structure is closer to the native conformation or whether the non-stabilized complexes are more anisotropic. It would be helpful to know whether the resolution around these regions different compared to other antibody-stabilized complexes.

--We appreciate the reviewer for this comment. As the reviewer pointed out, there are some GPCR-G_i complex structures were determined by the help of Fab scFv16, however it is not always necessary. Generally, we will try to avoid using antibodies if possible, as thus will be closer to its physiological condition. In the case of NMU2-G_i structure, stable complex of NmU-25-NMU2-G_{i1} could be obtained for structure determination without additional antibody fragment scFv16.

It needs to be specified that resolution around the N terminus of G α and the G β

subunit is enough for unambiguous modelling (see figure a below). It should also be noted that addition of Fabs has minimum influences on the overall architecture of GPCR–G protein complexes. Previous studies of G_i coupling of NTS₁ complexes determined with or without scFv16 have showed very similar conformation (Kato, Zhang et al. 2019, Zhang, Gui et al. 2021). In addition, superimposition of structures of GPCR– G_i complexes with or without scFv16 both showed similar conformational difference in G protein orientation as showed in Fig. 4a, b and Supplementary Fig. 8. In agreement with this, the structural alignment performed based on $G\alpha$ and $G\beta$ subunits of different GPCR– G_i structures showed no significant differences between the N terminus of $G\alpha$ and the $G\beta$ subunit (see figure b below), indicating that the relative conformation of this region was not affected by scFv16. So we believe that the differences observed in our complex was induced by the receptor itself and our complex structure reflect its real endogenous conformation. To make this clear, a sentence was added to the manuscript: “Similar conformational difference in G protein orientation is also observed by superimposition of other GPCR– G_i complexes without scFv16 addition, indicating that the specific conformational difference is induced by the receptor itself and NMU2– G_{i1} complex reflects its real endogenous conformation (Supplementary Fig. 8)” in line 249-253.

3. A related issue concerns whether there is something special about the NMUR2 and CB1 that allows the G_i N-termini to adopt this rotated conformation? It is pretty interesting that these two receptor- G_i complexes appear as outliers.

--We thank the reviewer for pointing this interesting observation out. We speculated that the specific rotated conformation of G protein in NMU2 and CB₁ receptor complexes represented the more energy-favorable orientation as extra interactions were observed in these complexes. The orientation difference of the G protein in CB₁ complex may be attributed to the more extended helix V of CB₁ where additional interactions (new polar interactions between R311^{5,75} from CB₁ and D337 and Q333 in $\alpha 5$ helix of $G\alpha_i$) with the $\alpha 5$ helix of $G\alpha_i$ were observed (Hua et al. 2020). For NMU2, two additional polar interactions (R333^{8,49} and P151^{34,50} from NMU2 and F354 and N347 in $\alpha 5$ helix of $G\alpha_i$) between the receptor and the $\alpha 5$ helix of $G\alpha_i$ were also observed (see figure 4d, 4e). The specific rotated conformations of NMU2 and CB₁ receptor complexes might be caused by these specific interactions between the

receptor and G protein that was not previously observed in other complexes. However, according to the published GPCR–G_i structures so far, it is difficult to summary a universal mechanism for these specific receptors upon binding to the same downstream effectors yet.

4. *It is curious why the authors used the GCGR-Gi complex to build their model but did not include the GCGR-Gi structure in their superposition with other Gi-coupled receptors. How do these N and C-termini positions differ in the GCGR-Gi complex relative to the NMUR2-Gi structure?*

--We apologize for this misunderstanding. We only used the DNG_{i1} part of GCGR–G_{i1} complex as a starting model to build the DNG_{i1} in NMU2–G_{i1} complex as both NMU2–G_{i1} and GCGR–G_{i1} complexes were used the similar dominant-negative G_{i1} (DNG_{i1}) for complex assembling. The overall structure of NMU2–G_{i1} was not build based on the GCGR–G_{i1} complex. We aligned NMU2–G_{i1} and GCGR–G_{i1} complexes based on receptor, N- and C-termini of G_{i1} appeared significant orientational difference as GCGR–G_{i1} complex showed similar rotation compared to NTS₁ complex (see figure below). However, in this manuscript we focused more on the G_{i1} difference in class A GPCRs, as the displacement of the intracellular tip of helix VI in the class B GPCRs was larger than any class A GPCRs–G_i structures and this indicate there might be some difference in the activation mechanisms between class A and class B GPCRs. To avoid over speculation, we hope that the reviewer would agree with us not to include superposition of GCGR–G_i structure in this manuscript.

5. *The authors appropriately included cholesterol hemisuccinate in their extraction, purification and imaging buffers. The protein concentration steps are appropriate and utilize cut-offs that should not concentrate empty micelles. The question is whether the receptor-G complex structures that the authors are comparing also include similar steps where cholesterol in particular is included at similar concentrations. Cholesterol and CHS are known to bind to regions surrounding TM7 and helix 8. Since these regions are critical for G protein coupling it is important to compare receptor complex stabilized in similar experimental conditions. It should also be noted that many of these differences might disappear if the receptor-G protein complexes were resolved in a lipid bilayer and not in detergent micelles.*

--We thank the reviewer for this comment. Among the complex structures discussed in

the manuscript, all except for β_2 AR-G_s complex, included cholesterol hemisuccinate with the concentration range of 0.000025%~0.001% during purification (NMU2-G_i, 0.0002%; NTS₁-G_i, 0.000075%; CXCR2-G_i, 0.00025%; CCK₁-G_i, 0.0002%; CB₁-G_i, 0.000025%; A₁-G_i, 0.001%) and concentrated in a similar manner. We agree with the reviewer that cholesterol molecules are very important for GPCR function, although no strong density map supported the cholesterol or CHS modelling surrounding the helices VII and VIII in these receptors yet. However, despite its important function, the influence of cholesterol on the receptor seems to be more relied on the stabilization of receptor while its influences on the conformation seems to be minimum. For example, the NTS₁-G protein complexes solved in lipid bilayer or in detergent micelles exhibit similar structural states of the complexes, indicating that the specific conformation of receptor-G_i complexes did not be affected by the lipids (Kato, Zhang et al. 2019, Zhang, Gui et al. 2021). On the contrary, although applied with the same concentration of cholesterol hemisuccinate, NMU2-G_i and CCK₁-G_i complexes revealed the conformational changes of G α as well. All these studies suggest that the differences of G protein orientation is receptor specific and the addition of cholesterol hemisuccinate has little effect on this. However, more structural information with higher resolution is required to illustrate the role of cholesterol and how CHS stabilizes or mediates GPCRs functions, which is beyond our manuscript.

6. In the description of the critical aspects for NMU25 C-terminus in efficacy of peptide ligand, the authors comment about the importance on the Asn25 amidation. It's important that the authors draw similarity to other monoamine receptors and their engagement of Asp3.32 or Glu3.33 for both binding affinity as well as agonist efficacy.

-- We appreciate the reviewer for this comment. The reviewer may refer to Asp^{3.32} or Glu^{3.32}, which is conserved in monoamine receptors and forms key interactions with the canonical primary amine of monoamine ligands. We superimposed the endogenous ligand-bound monoamine receptors with NMU2, the result showed that both of Asp^{3.32} in monoamine receptors and Glu^{3.33} in NMU2 formed polar interactions with receptors and played critical role in agonists' functionality (Xiao, Yan et al. 2021, Xu, Huang et al. 2021, Xu, Kaindl et al. 2021). However, as showed in the superimposition, the conserved Asp or Glu shifted one residue in helix III between the NmU and monoamine receptors, which led to the different position of amidation of Asn25 and amine of monoamine ligands.

We also compared NmU with all solved peptide receptors and found that D106^{3.33} in FPR2 and E124^{3.33} in ghrelin receptor formed hydrogen bonds with their corresponding ligands respectively and played significant roles in peptide agonist binding and agonist-induced receptor activity (Chen, Xiong et al. 2020, Wang, Guo et

al. 2021). For further explanation, we followed the reviewer's advice and added the statement “Mutants such as E127^{3.33}A and E102^{2.64}A caused a complete loss of the agonistic potency of NmU-25, indicating these residues play important roles in ligand recognition and receptor activation (Fig. 2f, Supplementary Fig. 6, Supplementary Table 2). This is similar with FPR2 and ghrelin receptors, where D106^{3.33} or E124^{3.33} forms hydrogen bonds with their corresponding ligands respectively and plays significant roles in peptide agonist binding and agonist-induced receptor activity^{29,30}. Similar recognition mechanism was also observed in monoamine receptors, which utilizing the acidic residues at 3.32 position to anchor the amine atom and activate the receptor²⁷” in line 141-151.

7. The authors note that agonist binding does not directly engage TM5, which is interesting. Many monoamine receptor engage TM5, as well as other TMs, that result in collapse of the core of the ligand binding, ultimately resulting in rotation and translation of TM6. It might be useful to include a superposition of TM core of NMUR2 with other active state GPCRs.

--We appreciate the reviewer for this comment. Indeed, all solved structures of monoamine receptors revealed that monoamine ligands bind with their receptors with conserved binding modes. For example, in D₁R and β_1 AR, the D^{3.32}-S^{5.42}-S^{5.46} motif of the receptor plays a significant role in maintain ligand binding and receptor activation. However, NmU receptors are peptide receptor and for peptide receptors, the activation of peptide receptors relies more to the key residue W^{6.48} than interaction with helix V. We followed the reviewer's comment and include a superposition of helical core of NMU2 with monoamine receptors and peptide-bound NTS₁, OX₂ and ghrelin receptor which share the high sequence similarity with NMU2 (34-41%) in Fig. 1b and Supplementary Fig. 5. The statement has been added: “Unlike the significant roles of helix V (D^{3.32}-S^{5.42}-S^{5.46} motif) of monoamine receptors in maintaining ligand binding and receptor activation, helix V of NMU2 plays minor role in ligand-mediated functions (Supplementary Fig. 5)²⁷” in line 118-120.

Reviewer #3 (Remarks to the Author):

1. While the data are certainly interesting, to be informative for the design of selective agonist for NMURs additional data is required to draw definitive conclusions on the structural basis of selectivity. While there is mutagenesis data in this manuscript on NMUR2 (introducing NMUR1 residues into NMUR2) that provide some support for key residues for selectivity, further evidence is warranted. For example, there is no structural, modelling or functional data provided for NMUR1 in the manuscript. Ideally structural data with selective ligands bound to each receptor, as well as non-selective ligands would be provided. However, In the absence of structure, minimally, if the identified residues are key to selectivity, it would be important to perform the reciprocal mutagenesis in NMUR1 and assess the selective NMUR2 agonist at these mutations (ie do they gain affinity over the WT NMUR1). Moreover, the selective NMUR1 agonist that is shown in this paper is not actually assessed at the introduced mutations in

NMUR2 to assess if the potency of this agonist is increased when the NMUR1 residues are introduced.

--We thank the reviewer for this comment. The sequence similarity between NMU1 and NMU2 is ~ 67%, which indicating that these two receptors might adopt very similar structural fold. The modelling of NMU1 based on NMU2 will not provide much extra information and thus it is not built in our original manuscript.

We followed the reviewer's advice and performed additional functional assays to prove our hypothesis about the subtype selectivity. The ligand preferences of different mutants were tested by both NMU1 and NMU2 selective ligands, and the results suggested that key residue did affect the ligand selectivity of NMURs. It was shown in our original manuscript that swapping the key residues in ligand binding pocket of NMU2 to NMU1 such as H43^{1.30} or N306^{7.32} greatly affected the potency of NMU2 selective ligands. Following the reviewer's suggestion, we tested the response of these mutants to NMU1 selective agonist, CPN267. The data showed that N306^{7.32}Q mutation could enhance the potency of CPN267 about 8-fold toward NMU2 compared with WT while H43^{1.30}E increased the E_{max} of CPN267 by 2-fold. These results suggest that different residues in 1.30 and 7.32 of NMU1 and NMU2 does play roles in selective ligand binding. This description above has been added: "The side chain difference in the ligand binding pocket might serve to the designation of NMU1 and NMU2 selective ligands. Replacement of H43^{1.30} or N306^{7.32} of NMU2 with glutamate or glutamine which are the corresponding residue in NMU1 respectively caused about 40-fold reduction of the potency of CPN 116 in stimulating NMU2 activation and A302^{7.28}H decreased the E_{max} of CPN116 dramatically (Supplementary Fig. 7c, Supplementary Table 2). On the contrary, these mutants exhibited better preference toward NMU1 ligands, as N306^{7.32}Q showed about 8-fold enhanced the potency compared with WT of CPN 267, a NMU1 selective ligand (Supplementary Fig. 7d, Supplementary Table 2). In the meanwhile, H43^{1.30}E also increased the E_{max} of CPN 267 by 2-fold but A302^{7.28}H had no improvement at all (Supplementary Fig. 7d, Supplementary Table 2). These results suggested that different residues in 1.30 and 7.32 of NMU1 and NMU2 might play a role in ligand selectivity and the results will provide the structural basis for future designation of selective drugs" in line 203-215.

2. Interestingly the authors identify a conformation of Gi2 that differs to the conformation observed in other Gi bound structures (relative to the receptor), including the closely related NTS1R. However, two conformations of Gi were observed in the NTSR1 structure. How does the conformation identified here compare to the second conformation observed for Gi bound to the NTSR1? Is there any evidence of other conformations of Gi in the NMUR2 dataset (ie only 900 000 particles of 3 million that were selected were used in the final reconstruction). Numerous studies have used 3D variance analysis to assess dynamics within complexes, and these reveal common rocking motions of the receptor relative to G protein that could lead to differences in consensus density maps depending on how the particles are selected. 3D variance analysis should be performed on this dataset to observe if the more common Gi conformation is present among the conformational space sampled at the time of

vitrification.

--We thank the reviewer for this comment. We compared two distinct conformations of G_i (canonical and non-canonical states) observed in the NTS_1 complexes (see figure a, b) (Kato, Zhang et al. 2019). The αN and $\alpha 5$ helix of G_i in the NMU2 complex lay between the canonical and non-canonical states of G_{i1} in the NTS_1 complexes (see figure a below). The canonical state of NTS_1 represented a full active state while the non-canonical state of NTS_1 displayed features of both active and inactive. On the contrary, in the non-canonical state of the NTS_1-G_{i1} complex, the motifs such as NPXXY indicate that the receptor lies in inactive states. Since the conformation of NMU2 display a full active state, we only compared canonical state of NTS_1-G_i with NMU2- G_{i1} complex in the original manuscript as they are in a comparable condition.

As for the particle number, a low threshold was set in Auto-picking in RELION at first to pick all possible particles to avoid missing and thus a lot of false positive particles were also selected and extracted. Auto-picked three million particles were confirmed by the 2D classification to remove false positive particles (see figure c). The best classes with one million particles were selected and subjected to later 3D classification. As the data processing pipeline shows, the one best looking class which contained 90% of one million particles was chosen for further refinement and polishing. So, there may not be another possible conformation. For clarity, we have revised Supplementary Fig. 2c.

a Structural superimpose of NMU2- G_{i1} , NTS₁- G_i (canonical state, 6OS9) and NTS₁- G_i (non-canonical state, 6OSA) complexes. Red arrows indicate the translocation αN helix of $G_{\alpha i}$. **b** Conformational change of $\alpha 5$ helix of $G_{\alpha i}$. **c** 2D classification result of three million particles.

To assess dynamics within NMU2- G_{i1} complex, 3D variance analysis was performed on the dataset of NMU2- G_{i1} complex using cryoSPARC as the reviewer suggested (movie 1, too large to be uploaded, will upload to google file dropbox upon reviewer' request). 3D variance analysis performed on NMU2- G_{i1} complex did show

rocking motions between NMU2 and G_{i1}, which was similar to previous study on CB₁-G_i-scFv16 complexes (Punjani and Fleet 2021). It is interesting that both NMU2 and CB₁ showed a different rotation of α N helix. The more common G_{i1} conformation is present among the conformational space sampled at the time of vitrification as the reviewer suggested, indicating the intrinsic flexibility and heterogeneity of GPCR for transient protein interactions. However, as the 3D classification and reconstruction only generated one electron density map with no other conformation classified, we believe that the NMU2-G_{i1} complex in our manuscript represented the most energy-favorable and physiological state. We added sentences in the manuscript to address the rock motion of G protein in relative to the receptor as: “Furthermore, 3D variance analysis was performed on the dataset of NMU2-G_{i1} complex using cryoSPARC to assess dynamics within NMU2-G_{i1} complex (movie 1, too large to be uploaded, will upload to google file dropbox upon reviewer’ request). 3D variance analysis did show rocking motions on an axis parallel to the membrane between NMU2 and G_{i1}, which was similar to previous study on CB₁-G_i-scFv16 complexes⁴⁶. It is interesting that NMU2 and CB₁ showed a different rotation of α N helix in comparison with other GPCR-G_i complexes. These results suggested the intrinsic flexibility and heterogeneity of GPCR for transient protein interactions” in line 325-332.

3. It is also interesting that the authors did not see density in the cryo-EM map for helix 8. Why do the authors think this is the case? Do they think that the receptor lacks helix 8, or that it becomes unstructured in the presence of Gi? It would be interesting to assess this in the 3D multivariate analysis to see if there is any evidence within the particle stack of the presence of H8. It is also important to note that the structure is in detergent and therefore H8, the α N and β g subunits of the G protein might be influenced differently in detergent vs lipid, ie the different Gi conformation could be an artefact of detergent extraction.

--We appreciate the reviewer for this comment. It is believed that upon activation, the intracellular side of the helix bundle open up to adopt G protein, which will in turn increases the flexibility of helix VIII. This also aligns well with the 3D variance analysis, which showed significant motions of helix VIII (movie 2, too large to be uploaded, will upload to google file dropbox upon reviewer’ request). Thus, after averaging, the cryo-EM density map processed by RELION showed no clear density of this helix and it is

a Cryo-EM density map and fitted model of the C terminus of NMU2. **b** Structural alignment of canonical state of NTS₁-G_{i1} complexes embedded in detergent micelles (6OS9) or lipid bilayer (7L0P). Hydrogen bond is indicated by green dash line.

not possible to trace it based on the cryo-EM map (see figure a below). We have clarified the description about helix VIII in line 88-89 and 260-263.

As for the influence of lipids and detergents on the receptor and G protein, we agree with the reviewer that receptor and G protein might behave differently upon detergents and lipids. However, we believe that the conformational difference we observed in the manuscript is more likely to be receptor specific instead of caused by artifact of detergent extraction as we stated in response 4 to Reviewer #2.

4. Can the authors clarify why they selected *Gi1*? This receptor couples to other *Gi* subunits as well as *Gq*. Given all the functional data was performed using an IP1 assay (downstream of *Gq*), why was *Gq* not used in the structural studies?

--We thank the reviewer for this comment. It has been stated that NmU-induced activation of NMU1 and NMU2 resulted in coupling of the receptor to *Gi* and *Gq* (Brighton, Szekeres et al. 2004). Moreover, there were some literatures verified that

NMU2 transduced more potent *Gi* signals than *Gq* signals (Aiyar, Disa et al. 2004, Brighton, Szekeres et al. 2004, Hsu and Luo 2007). To clarify the G protein signaling NMU2 conducted, we performed TRUPATH assay by testing the NmU-25-induced potency on *Gi1*, *Gq* and *Gi11* signaling pathways, and similar to previous data, our results showed that NmU-25 had the highest potency by coupling *Gi1*. In agreement with functional results, comparison the purified NmU-25-NMU2-*Gi/q/11* complex formed in vitro using analytical size-exclusion chromatography showed the most promising complex behavior and thus *Gi1* was chosen for structural studies. A sentence was added to the method section to facilitate the readers as "Comparison the NmU-25-NMU2-*Gi/q/11* complex formation in vitro using analytical size-exclusion chromatography showed the most promising complex behavior and thus *Gi1* was chosen for structural studies."

5. Mutagenesis was used to validate the interactions between ligand and receptor. All changes in potency of the mutant receptors have been interpreted as reduced interaction of the ligands with the receptor. However, in a large number of the mutants, receptor expression was significantly lower than WT. Given a functional assay (which is influenced by receptor expression levels) was used to validate the mutants, how can the authors distinguish whether the effects on potency are related to the lower expression or reduced interactions of the ligands with the receptor?

--We appreciate the reviewer for this comment. Five mutants in TRUPATH assay

a G protein types test of NmU-25-induced NMU2 activation by TRUPATH sensor. Data are shown as mean ± SEM from at least three independent experiments performed in triplicate. **b** Size-exclusion chromatography of NmU-25-NMU2-*Gi/q/11* complexes test.

(H185F, F284A, R288A, H309A and H309F) and one mutate (R288A) in IP1 accumulation assay presented lower expression level (less than 40% of wild-type). We have tried to optimize the multiplicity of infection ratio of these mutates and G protein, but no improvement of expression level was observed. All of these mutants with lower expression level both decreased the E_{max} of the agonist as well, especially for H185F, R288A, H309A and H309F. We presumed that these mutants might impair the receptor trafficking to the cell surface or receptor stability and we had clarified in the manuscript as the reviewer suggested: “H185^{4.64}F and R288^{6.55}A greatly reduced the level of receptor expression, suggesting that these residues might play critical roles in mediating the stability of ligand-binding pocket which is important for ligand recognition (Supplementary Table 2)” in line 148-151, “These mutants also showed lower expression level in comparison of wild type receptor, which might also contribute to the weaker binding of corresponding ligands.” in line 158-160, “E105^{2.64}A also decreased the E_{max} of NmU-25 by 50%, which might be caused by weakened binding affinity as well as the reduction of the receptor expression level (Supplementary Table 2).” in line 168-170.

6. *Figure 2 – panels e and f are labelled incorrectly.*

--We thank the reviewer for this comment. We have corrected panel labels of Fig. 2.

7. *Line 168-169 – 100-fold selectivity for NMUR2 – should this read over NMUR2????*

--We thank the reviewer for this comment. The word “for” has been corrected to “over” as suggested.

8. *Supp Fig 6c – should this be [CCPN116] on the x axis, not [NMU-25]?*

--We thank the reviewer for this comment. The x axis of Supplementary Fig. 6c has been corrected.

9. *Line 204 references a TRUPATH assay, yet none of the figures referenced show TRUPATH data.*

--For clarity, we have omitted the discussion about IP accumulation and TRUPATH assay followed the suggestion of reviewer #4.

10. *There is no method reported for the flow cytometry assay used to measure receptor expression and there are a large number of grammatical errors throughout the manuscript that will need to be addressed.*

--We reported the procedure of the flow cytometry assay in line 432-435. Grammatical errors have corrected in the manuscript carefully. To avoid confusion, the method was separated into two paragraph to explain the expression and activity assays.

Reviewer #4 (Remarks to the Author):

1. *There are a number of typographical and grammatical errors that require attention.*

Furthermore, some aspects require a little work to improve accuracy and to help the reader a little more. This could include more helpful statements about the roles of NmU ie. where possible, be explicit in the roles rather than simply stating regulates, elicits etc. The figures are very nice and present the information clearly. I have one or two queries about the colour references in the figure legends, which themselves would also benefit from a little editing.

--We appreciate the reviewer's comment. Followed the reviewer's suggestion, we corrected typographical and grammatical errors carefully and expanded the description about the roles of NmU. The statement "NmU-25 and NmU-33 are ubiquitously distributed through the body and implicated in the regulation of smooth muscle contraction, energy balance, feeding behavior, pronociception and tumorigenesis" was added in line 48-50 and the color references have been checked carefully.

2. *It would be prudent to use the IUPHAR recommended abbreviations for the receptors and*

ligands: <https://www.guidetopharmacology.org/GRAC/FamilyDisplayForward?familyId=42>

Ensure abbreviations defined at first use eg. NTSR1, β 2AR.

--We thank the reviewer for this comment. According to the IUPHAR recommended abbreviations, we changed the NMU25 to NmU-25, NMUR2 to NMU2, NTSR1 to NTS₁, CB1 to CB₁, A1R to A₁, β 2AR to β ₂AR and other clerical error to avoid misleading.

3. *Accession codes are missing.*

-- We thank the reviewer for this reminder. We have submitted the model and the cryo-EM map to Protein Data Bank and Electron Microscopy Data Bank and accession codes will be added during revision stage.

4. *Some care needs to be taken with the terminology around changes in ligand potency associated with mutagenesis. For example, breaking the salt-bridge between R22 of the ligand and E105 by mutating to alanine, is said to decrease the EC₅₀ about 50-fold. This mutation increases the EC₅₀ (37 nM in WT to 1854 nM in E105A). The pEC₅₀ and agonist potency are reduced. Similarly, the mutations N109A, F305A and A302H increase rather than decrease the EC₅₀. Additionally, when discussing CPN116 and CPN267, the authors switch between potency and affinity. As affinity has not been measured, it is not possible to directly attribute changes in potency to changes in affinity; the data could certainly suggest or support this (and should be phrased appropriately). Replacement of H43 or N306 do not decrease the EC₅₀; they increase the EC₅₀ and lower potency.*

--We thank the reviewer for this comment. We corrected these descriptions for previous discussion about the result of functional assay to avoid unnecessary mis-leading: "Breaking the salt bridge between R22 and E105^{2,64} compromised the agonist potency of NmU-25 about 50-fold" in line 164-165, "which was further supported by weakening the potency of NmU-25 for these mutants" in line 182-183, "caused about 40-fold

reduction of the potency of CPN 116 in stimulating NMU2 activation” in line 206-207.

When discussing CPN 116 and CPN 267, we cited previous studies for demonstrating the characteristics of these two ligands. According to the researches and reviewer’s comment, affinity has been changed to potency for clarity in line 189.

5. NMU2 is clearly able to couple to Gi. There is functional experimental evidence to support this and it is not uncommon for Gs- and Gq-coupled receptors to show such promiscuity. However, given that NMU2 is preferentially a Gq-coupled receptor, some comment about the choice of G-protein in the present study would be warranted, along with at least some discussion of the relevance and any potential consequence of Gq coupling in relation to the present findings. For example, given the C-terminus of the G-protein is widely considered to be responsible for driving recognition by the receptor, are there any consequences of using Gi rather than Gq to determine the binding and activation mechanisms? Are different conformations required for Gi and Gq coupling?

--We thank the reviewer for this comment. As the response to the forth comment of Reviewer #3, the data presented that NmU-25 had the highest potency upon activating NMU2 by coupling Gi1 and promising NmU-25–NMU2–Gi complex in vitro assembling with higher monomer : aggregation ratio was chosen for structural studies.

Up to now, eight GPCRs–Gq/11 complexes (M1–G11, 6OIJ; 5-HT2A–Gq, 6WHA; H1–Gq, 7DFL; CCKAR–Gq, 7EZM; CCKBR–Gq, 7F8W; ghrelin receptor–Gq, 7F9Y; B1R–Gq, 7F2O; GPR139–miniGs/q, 7VUH) have been reported (Maeda, Qu et al. 2019, Kim, Che et al. 2020, Liu, Yang et al. 2021, Wang, Guo et al. 2021, Xia, Wang et al. 2021, Yin, Ye et al. 2021, Zhang, He et al. 2021, Zhou, Daver et al. 2021). Structural alignment of these Gq/11-coupled complexes with NMU2–Gi1 complex revealed nearly identical conformations of receptors but diverse G protein binding modes as well (see figure below). We summarized the key interactions between class A GPCRs and Gq/11 that play critical roles in maintaining Gq/11 signaling. The interaction interface between GPCRs and G protein is mainly mediated by the $\alpha 5$ helix of G α subunit. As the C-terminal “wavy hook” of Gq/11, the conserved C terminus residue, L358^{H5.25}, formed conserved hydrophobic interactions with V/A^{6.33} and L^{6.37}. Two non-conserved residues of Gq, Y356^{H5.23} and N357^{H5.24}, form polar interactions with D^{3.49}, R^{3.50}, N/S^{8.47} or N/R^{8.49} of the receptor which are important for Gq/11 protein coupling. A highly conserved hydrophobic residues (I, L or F) and a hydrophilic residue in ICL2^{45.51} and ICL2^{45.54} also form critical interactions with Gq/11. Basically, the residue of NMU2 meets the needs of the above interactions. But beyond that, Gq/11 protein binding exhibited specific interactions for different receptors as well. For example, in M1–G11 complex, the C terminus of M1 after helix VIII that extend into a groove formed by the Ras domain of G α and the G β . A similar C-terminal extension has not been observed in other GPCR–G protein complexes reported so far. In the CCK1–Gq complex, the ICL3 of the receptor interacted with the hydrophobic patch tightly formed by G α_q subunit specifically, which were critical to CCKAR–Gq coupling. We propose the potential consequence of Gq coupling of NMU2 is that the residue of NMU2 meets the needs of the Gq coupling, but different conformations required for Gi and Gq coupling of NMU2 need further structural information. We have discussed potential consequence

of G_q coupling followed reviewer's suggestion in revised version: "Besides G_i signaling pathway, NMU2 was reported to couple G_q for signaling transduction as well²⁸. The key interactions between class A GPCRs and G_{q/11} that play critical roles in maintaining G_{q/11} signaling are mainly mediated by several conserved interactions. As the C-terminal "wavy hook" of G_{q/11}, the conserved C terminus residue, L358^{H5.25}, forms hydrophobic interactions with V/A^{6.33} and L^{6.37} of the receptor. Two non-conserved residues of G_q, Y356^{H5.23} and N357^{H5.24}, form polar interactions with D^{3.49}, R^{3.50}, N/S^{8.47} or N/R^{8.49} of the receptor. A highly conserved hydrophobic residues (I, L or F) and a hydrophilic residue in ICL2^{45.51} and ICL2^{45.54} also mediate interactions with G_{q/11}³⁸⁻⁴⁵. The above residues all appear to be conserved in NMU2, indicating that NMU2 might bind to G_q in a similar manner as previously reported." in line 288-298.

6. *Although the determined structures and the bulk of the pharmacology (using the TRUPATH assay) have focused on Gi-coupling, the data around the antagonism of R-PSOP addresses IP accumulation and therefore Gq-coupling. Are there any issues around switching between different cellular outcomes with different transduction pathways?*

--We appreciate the reviewer for this comment. TRUPATH assay is a direct and sensitive method for verification of receptor activation by monitoring the fluorescent signal changes of agonist-induced G α and G $\beta\gamma$ departure. When measuring the antagonist effect, a certain concentration of R-PSOP or NmU-25 in advance and then the antagonists were added to inhibit receptor. In this case, part of G α and G $\beta\gamma$ was already departed and they could not form the heterotrimer instantly upon antagonist inhibition. In this case, it is very hard to obtain dose-dependent response of antagonist. So we applied IP accumulation for further measure antagonist effects. It should be stated that the results of these two different assays align with each other pretty well in our previous assays, so we believe that these data are comparable with each other.

7. *One of the difficulties when assessing the impact of mutation on the coupling/function of receptors is interpreting the potential confounding effects of changes in cell surface expression. It was reassuring to note that the authors have assessed expression but differences in expression were not addressed as part of the consideration of the effect of mutation on function. For example, in the data from the TRUPATH assay, there was a fairly notable positive relationship between the expression of the mutant receptor (compared to wild-type) and the E_{max} values. Is there any possibility that one or more of the mutants impacted on, for example, receptor stability and/or receptor trafficking? Over what range of expression levels are wild-type responses consistent (ie. similar in EC₅₀ and E_{max}) and does this include the range covered by the expression of the mutant receptors? Do these points merit discussion? As specific examples, it is suggested that mutations of either N109, A302 or F305 decrease agonist potency. There is a presumption that this is due to reductions in ligand affinity but this has not been determined. Changes in EC₅₀ may result from changes in affinity, intrinsic efficacy and receptor expression levels.*

--We appreciate the reviewer for this comment. As the response to the fifth comment of

Reviewer #3, we had clarified in the manuscript as reviewer suggested: “H185^{4.64}F and R288^{6.55}A greatly reduced the level of receptor expression, suggesting that these residues might play critical roles in mediating the stability of ligand-binding pocket which is important for ligand recognition (Supplementary Table 2)” in line 148-152, “These mutants also showed lower expression level in comparison of wild type receptor, which might also contribute to the weaker binding of corresponding ligands.” in line 158-160, “E105^{2.64}A also decreased the E_{max} of NmU-25 by 50%, which might be caused by weakened binding affinity as well as the reduction of the receptor expression level (Supplementary Table 2).” in line 168-170.

But there may not be a definite relationship between the expression levels of the mutant receptors (compared to wild-type) and the E_{max} or potency values. For F291A, it only had a 66% expression level with a span of 127% but a reduced potency compared with wild-type. For F305A, it had a comparable expression level (97%) but with lower E_{max} (83%) and potency (33-fold reduction). It could not be clarified that the reduction of the potency of the agonist is a consequence of reduced ligand affinity or a reduction in ligand efficacy, which may need further ligand binding verification. In our manuscript, disrupting these interactions between receptor and ligands compromised the ligands’ potency and E_{max} to varying degrees, which is consistent with our structural information.

8. For some of the mutants, there were significant effects on agonist potency in the TRUPATH assay. Again, is it possible that any of the shift in EC50 values were a consequence of alterations in expression? Further, where mutation reduced agonist potency without impacting receptor expression (e.g.H43E and N100A), would the data suggest that this may be a consequence of reduced ligand affinity or a reduction in ligand efficacy? Perhaps a little more discussion and clarity would add to the manuscript.

--We thank the reviewer for this comment. The system we used for activity assays is overexpression system, and the NMU2 receptor showed a very high expression, and the actual expression level even for the mutants with lower expression levels (i.e. ~30%) in comparison to the wild type receptor still relatively high. The activation curves of all the mutants are in a very good quality with good reproducibility, which were provided in the supplementary information. So we believe that the affinities determined of these mutant more likely reflect the real data. However, one could not rule out the possibility that the expression level also influence the affinity to some extent, so we have clarified this question by suggesting that the expression levels were also different and the affinity and the E_{max} measured might also affected by the lower level. For more detail please see response 7 to this Reviewer.

9. In supplementary figures 5 and 6, it perhaps should be stated that the lowest concentration of ligand used was either zero or -11 (log10 M). I appreciate the difficulties of expressing zero on this scale.

-- We thank the reviewer for this comment. Actually, the lowest concentration of the ligand used in TRUPATH and IP accumulation assay was -11 (log10 M) and we

described this detail in the part of method. For clarity, we followed the reviewer's suggestion and added statement in the caption of supplementary figures 5 (6) and 6 (7) and updated the supplementary figures 5 (6) and 6 (7).

10. Would it be worth commenting a little further on why the specific receptors have been selected for more detailed comparisons (NTSR, OX2R) and the significance of these?

--We appreciate the reviewer for this comment. Both NTS₁ and OX₂ are the structures determined up to date which share the highest sequence similarity with NMU2, and these two receptors happened to be peptide binding receptors as NMU2. So we selected these two receptors for detailed comparisons. Followed the reviewer's suggestion, the statement "The extracellular part of NMU2 is widely opened to accommodate peptide ligand compared with other solved peptide-bound receptors which share the high sequence similarity with NMU2 (Fig. 1b)" has been added in line 94-95.

11. Whilst the manuscript has focused on NmU-25 as the endogenous ligand, highlighting the consistency of the C-terminus of this and other peptide ligands, do the data allow any discussion of potential roles for the N terminus and in particular differences here between the endogenous ligands (ie. NmU and NmS)?

--We appreciate the reviewer for this comment. Both NmU-25 and NmS-33 had equal binding affinity to NMU1 and NMU2 but NmS-33 showed a higher binding affinity for NMU2 than NmU-25 (Mori, Miyazato et al. 2005). From the perspective of structural biology, the N terminus of NmU-25 only forms minor hydrophobic interactions with NMU2, so we speculate that the extra N terminal residues of NmS-33 such as arginine or glutamine might form more interactions with NMU2 for higher binding affinity. But without more structural information, this view is difficult to be proved. For further discussion, the statement "It has been known that NmS-33 showed a higher binding affinity for NMU2 over NmU-25²⁸. This might be benefitted by potentially more interactions were formed between the hydrophilic residues such as arginine or glutamine in the extra N terminus of NmS-33 and receptor ECL2, providing a strengthened interaction network between N terminus of peptide and the receptor. However, more data is needed before a clear conclusion was made." has been added in line 130-135.

12. Generally, the Discussion is somewhat limited and would benefit from a wider consideration of the findings.

--We followed the reviewer's suggestion and extended the discussion section at page 10-11.

13. Might it be useful for Supplementary Figure 4 to contain an indication of the location of the TMDs, ECLs and ICLs?

--We have followed reviewer's advice to improve Supplementary Fig. 4.

REVIEWERS' COMMENTS

Reviewer #1 (Remarks to the Author):

Regarding my comments, the manuscript has been significantly improved and can be accepted in the current form.

Reviewer #2 (Remarks to the Author):

The revised manuscript by Zhao et al includes several modifications in response to this and other reviewers. An improved discussion, the addition of new data and a more thorough comparison with other GPCR-G protein structures improve the manuscript in general.

There are some remaining issues that need to be addressed:

Comments:

In their rebuttal the authors claim that the specific conformations of the lipid-bound versus detergent-stabilized NTS-Gi complexes are similar, implying that the lipids play little role. The authors should revisit this statement as careful examination of the two structures reveal significant differences. The most notable differences are between the receptor G protein interface, including the alphaN of Galphai and the interaction with ICL2. In lipid, ICL2 appears to be closer to the lipid face. This is likely influenced by the interaction of both ICL2 and the alphaN of Galphai with the phospholipid headgroups in the bilayer. In addition, the angle of attack of the C-term of Galphai is slightly different (approx. 12-15 degrees) in lipid, with respect to the receptor. In the detergent structure this rotation allows the alpha4-beta5 loop of Galphai to be closer to helix 8 of NTS1R, similar to what seems to be observed with NMUR1/2. Such an interaction is not possible in a phospholipid bilayer. So it is clear that the lipid plays a significant role in both receptor and G protein conformation.

In line 51 the authors refer to two intrinsic GPCRs. Do the authors mean endogenous GPCRs or intrinsic membrane proteins or something else ?

In the results section the authors should remind readers that they are using mutant version of Galphai. The mutant contains 5 mutations which effect nucleotide exchange, Mg²⁺ coordination and Switch II coordination (and therefore Gbetagamma interaction).

In lines 170-172 the authors discuss E105(E2.64) and the decreased Emax. This should NOT be caused by the weakened affinity but could be affected by expression levels.

In line 262-267 the authors make reference to an interaction between R333(R8.49) and the C-term of Galphai. This is unlikely to occur in a phospholipid bilayer since helix 8 would be buried in the bilayer, with the Arg side chains likely interacting directly with the phospholipid head group. The discussion of the positively-charged helix 8 following this sentence will likely have to be modified for this reason (see above discussion the lipid vs detergent structures).

Minor Comments:

There are several grammatical errors (too many to list here) that need to be addressed. These are likely to be picked up by the editorial staff.

Line 186 'interacted' should read 'interacting'

Line 251 should read 'infected' instead of 'transfected'.

Reviewer #3 (Remarks to the Author):

The authors have adequately addressed all of my comments, and the revised manuscript is substantially improved. The revised manuscript makes a valuable contribution to the field and I would recommend publication.

Reviewer #4 (Remarks to the Author):

My general comments about the paper made in the initial review still stand.

There are still a substantial number of typographical/grammatical errors that would, no doubt, be picked-up during the process.

Some suggested corrections/alterations for the Abstract, Introduction, Results and Discussion are given below. There are grammatical corrections required in other sections including the figure legends.

Below also contains one or two comments/questions.

Line 36 high – highly

Line 37 degree rotation of Gi protein is relevant to NMU2 compared with (?)

Line 48 NmU-33 – NmS-33. Also, this line now perhaps provides misleading information about the distribution and roles of NmS-33. I would certainly argue that it is not ubiquitously expressed as indicated and may not play the full range of roles listed that NmU-25 may.

Line 58 – the practical – practical

Line 59 NMUs – NMU

Line 63 activation of NMU2 would dramatically decrease body weight and food - activation of NMU2 dramatically decreases body weight and food

Line 64 intake in mice while in contrast inhibition of NMU2 would promote weight gain and - intake in mice while in contrast inhibition of NMU2 promotes weight gain and (?)

Line 66 is lagged behind partially due to the lack of structural information – has been limited, partially due to the lack of structural information.

Line 68 complex bound to endogenous peptide NmU - complex bound to the endogenous peptide NmU

Line 70 which is prerequisite for ligand recognition - which is a prerequisite for ligand recognition

Line 75-79 This lacks a little clarity. the flexible C-terminal residues Q356 to T415 of NMU2 were replaced by a twin-strep affinity tag and a hemagglutinin (HA) signal peptide followed by a flag tag was introduced at N terminus to improve expression

Line 82 (named TRUPATH) – (TRUPATH)

Line 87 unambiguously with clear and strong density map (Supplementary Fig. 2g) - unambiguously with a clear and strong density map (Supplementary Fig. 2g).

Lines 88-90 The overall structure of NMU2 possesses canonical seven transmembrane helical domain compared with other solved peptide receptors of class A GPCRs with helix VIII unmodelled due to its flexibility upon activation - The overall structure of NMU2 possesses a canonical seven transmembrane helical domain similar to other solved peptide receptors of class A GPCRs with helix VIII unmodelled due to its flexibility upon activation (?)

Lines 94-95 The extracellular part of NMU2 is widely opened to accommodate peptide ligand is widely opened to accommodate peptide ligand compared with other solved peptide-bound receptors which share the high sequence similarity with NMU2 – To accommodate the peptide ligand, the extracellular part of NMU2 is more widely opened compared to other solved peptide-bound receptors. (?)

Lines 96-97 NMU2 adopts full active conformation in intracellular side stabilised - NMU2 adopts a fully active conformation at the intracellular side, stabilised

Line 101 indicate that NMU2 is in active state - indicate that NMU2 is in an active state

Line 102 is in active-like conformation - is in an active-like conformation

Line 103 the rearrangement of “P5.50-I3.40-F6.44” motif - the rearrangement of the P5.50-I3.40-F6.44 motif

Line 106 arousing the rearrangement of – causing the rearrangement of

Line 107 of $\alpha 5$ helix of G protein - of the $\alpha 5$ helix of the G protein

Line 108 to form a hydrogen network - to form a hydrogen network

Line 113 peptides with various forms of length share the - peptides with various lengths share the

Line 115 which is crucial for ligand’s agonistic - which is crucial for the ligand’s agonistic

Line 117 structure with the conserved C terminus penetrating deeply in the entire orthostatic - structure with the conserved C terminus, penetrating deeply in the entire orthostatic

Line 121 of NMU2 plays minor role in ligand - of NMU2 plays a minor role in ligand

Line 123 contributes a large portion of interaction - contributes a large portion of the interaction

Line 126-127 to accommodate in the space between ECL1, ECL2 - to facilitate accommodation in the space between ECL1, ECL2

Line 128 For example, Y18 packs against with the main chain of ECL3 by hydrophobic interactions while - For example, Y18 packs against the main chain of ECL3 by hydrophobic interactions, while

Line 130 Besides, P199 - In addition, P199

Line 131 It has been known that NmS-33 showed a higher binding affinity for NMU2 overNmU-2528. This might be benefitted by potentially more interactions were formed between the hydrophilic residues such as arginine or glutamine in the extra N terminus of NmS-33 and receptor ECL2, providing a strengthened interaction network between N terminus of peptide and the receptor. However, more data is needed before a clear conclusion was made. – There has been some indication that NmS-33 may have a higher binding affinity for NMU2 compared to NmU-2528. This might result from potentially more interactions between the hydrophilic residues such as arginine or glutamine in the extended N terminus of NmS-33 and receptor ECL2, providing a strengthened interaction network between the N terminus of the peptide and the receptor. However, more data are needed before a clear conclusion can be made.

Note that although this paper is often cited, there is little confirmatory data. In most studies, no real differences have been shown. Furthermore, the potencies of these two peptide ligands tend to be equivalent suggesting that unless NmS has a reduced intrinsic efficacy that precisely offsets a higher affinity, that this may not be the case.

Line 138 As for the C terminus, last four residues of NmU-25 – The last four C-terminal residues of NmU25

Line 143 caused a complete loss of the agonistic potency – have a complete loss of potency

Line 147 forms – form

Line 147 respectively and plays - respectively and play

Line 149 Similar recognition mechanism was also observed in monoamine receptors, which - Similar recognition mechanisms are also observed in monoamine receptors, which

Line 150 utilizing the acidic residues - utilize the acidic residues

Line 152-153 might play critical roles in mediating the stability of ligand-binding pocket which is - might play critical roles in mediating the stability of the ligand-binding pocket, which is

Line 159 alanine impairs the receptor activation strikingly – strikingly impairs receptor activation

Line 160-162 These mutants also showed lower expression level in comparison of wild type receptor, which might also contribute to the weaker binding of corresponding ligands. - These mutants also showed lower expression levels in comparison of wild type receptor, which might also contribute to the weaker binding of corresponding ligands.

There needs to be some caution in such statements. Receptor-ligand affinity has not been measured. The authors have measured ligand potency. This is a reflection of affinity, efficacy and receptor expression. It is an assumption that it is affinity that is affected by mutagenesis. This may fit with the model but needs to be made clear (ie. that affinity has not been measured).

Line 164 alanine would result in inactive peptides - alanine results in inactive peptides

Lines 170-172 E1052.64A also decreased the Emax of NmU-25 by 50%, which might be caused by weakened binding affinity as well as the reduction of the receptor expression level (Supplementary Table 2).

A reduction in Emax is not consistent with a reduced binding affinity. Provided that the efficacy of a ligand is not reduced, a reduction in binding affinity would simply mean that a greater concentration of ligand would be required for the Emax – but it would be reached. A reduction in Emax can occur when efficacy is reduced (receptor mutation could cause this – the receptor is less well-coupled to signalling mechanisms) and/or receptor expression levels fall below those required for the maximal response (ie. the receptor reserve is removed).

Line 172-173 These structural evidences suggest – This structural evidence suggests

Line 173 between this hydrophilic portion and the receptor - between this hydrophilic portion of the ligand and the receptor

Line 174 mediating agonistic functionality of NmU peptides - mediating agonism of NmU peptides

Line 175 alignment presents that these - alignment demonstrates that these

Line 178 pack against with each other - pack against each other

Line 183 (residue resides in NMU1) – (residue of NMU1) (?)

Line 184 which was further supported by weakening the potency of NmU-25 for - which was further supported by the reduced potency of NmU-25 for

Line 185-186 Residues interacted with - Residues that interact with

Line 187 MNU2 – NMU2

Line 190 CPN 116 and CPN 267 were - CPN 116 and CPN 267, were

Line 191 (residue 19-25) - (residues 19-25)

Line 191 (residue 19-25) with different selective characteristics - (residues 19-25). These had different selectivity characteristics

Line 193 CPN267 owned high agonistic activity for NMU1 - CPN267 showed high agonistic activity for NMU1

Line 196 position of 20 and 21. - position of residues 20 and 21.

Line 199 NMU2 complex reflects that residues interacted with L20 and F21 of - NMU2 complex indicates that residues that interact with L20 and F21 of

Line 201 residues affect the ligand selectivity - residues affect ligand selectivity

Line 202 NMU2 with relative residues in NMU1 - NMU2 with the respective residues of NMU1 (?)

Line 203 Substitution the H43 - Substitution of H43

Line 204 with bulky residues tryptophan - with the bulky tryptophan residue

Line 206-208 The side chain difference in the ligand binding pocket might serve to the designation of NMU1 and NMU2 selective ligands. - The side chain difference in the ligand binding pockets may underlie the receptor-selectivity of ligands. (?)

Line 209 glutamine which are the corresponding residue in NMU1 respectively – glutamine respectively, which are the corresponding residues of NMU1,

Line 210 caused about 40-fold reduction - caused about a 40-fold reduction

Lines 210-211 CPN 116 in stimulating NMU2 activation and A302 - CPN 116 in stimulating NMU2 signalling. A302

Lines 213-214 showed about 8-fold enhanced the potency compared with WT of CPN 267, a NMU1 selective ligand - showed about 8-fold enhanced potency with CPN 267, a NMU1 selective ligand, compared to the WT receptor

Line 215 In the meanwhile, - In addition,

Line 216 had no improvement at all – had no effect

Line 217 These results suggested that - These results suggest that

Lines 218-219 and the results will provide the structural basis for future designation of selective drugs. - and provide a structural basis for the future design of receptor-selective drugs.

Line 220 binding mode of non-peptide antagonist - binding mode of a non-peptide antagonist

Line 225 stabilized through the polar interactions - stabilized through polar interactions

Lines 226-227 are bordered by hydrophobic - are bordered by a hydrophobic

Line 227 F2846.51, F3167.42 - F2846.51 and F3167.42

Line 231 we carried out functional experiment by an inositol phosphate (IP) accumulation assay to test the effects of these residues involved in R-PSOP binding. - we carried out an inositol phosphate (IP) accumulation assay to test the effects of these residues involved in R-PSOP binding.

It might be worth highlighting that this assay is likely to measure predominantly Gq-coupling (there is no evidence of eg. pertussis toxin-sensitive (Gi-mediated) inositol phosphate accumulation). How does this relate to the structural data on Gi-coupling?

Lines 235-237 which might cause by lowering the expression level of the receptor greatly (Supplementary Fig. 7e, Supplementary Table 3). - which may be a consequence of the dramatically reduced expression level (Supplementary Fig. 7e, Supplementary Table 3).

Line 239 V is not conserved between NMU1 and NMU2 - V are not conserved between NMU1 and NMU2

Line 242 with leucine or cysteine could decrease the antagonistic - with leucine or cysteine decreased the antagonistic

Line 245 ligand selectivity for non-peptide antagonist. - ligand selectivity for the non-peptide antagonist. or ligand selectivity for non-peptide antagonists. (depending on whether this is a specific statement relating to R-PSOP or non-peptide antagonists generally).

Line 251 in NMU2-Gi1 - in the NMU2-Gi1

Line 251 to that in hNTS - to that in the hNTS

Lines 252-253 (Fig. 4a). $\alpha 5$ helix is tilted with different degrees compared with $G\alpha i 1$ in other GPCR-Gi complex (Fig. 4b). - $\alpha 5$ helix is tilted to a different extent compared with $G\alpha i 1$ in other GPCR-Gi complexes (Fig. 4b).

Line 253 Similar conformational difference in G – A similar conformational difference in G

Line 256 induced by the receptor itself and NMU2 - induced by the receptor itself and that the NMU2

Line 262 and NMU2 compared to other $G\alpha i 1$ complexes - and NMU2 compared to other GPCR- $G\alpha i 1$ complexes (?)

Line 265 forms a salt bridge with C-terminal - forming a salt bridge with the C-terminal

Line 266 there is no other - there are no other

Line 268 conserved hydrogen bond between C-terminal carboxyl of $G\alpha$ - conserved hydrogen bond between the C-terminal carboxyl of $G\alpha$

Lines 270-271 coupled with Gi/o protein reveals that only a small number of receptors have a positively - coupled with the Gi/o protein reveals that only a small number of receptors have positively

Line 274 with G protein - with the G protein

Line 279 ICL2 played - ICL2 plays

Line 281 receptor own a bulky - receptor have a bulky

Line 284 of αN helix - of the αN helix

Line 285 of receptor and G - of the receptor and G

Line 286 Similarly, the hNTS - Similarly, hNTS

Lines 286-287 it also mediate weaker - it also mediates weaker

Line 287 of hNTS - of the hNTS

Line 288 compared with β 2AR - compared with the β 2AR

Lines 290-291 even though with same aromatic amino acid present - even though the same aromatic amino acid is present

Lines 292-293 Besides Gi signaling pathway, NMU2 was reported to couple Gq for signaling 292 transduction as well - Besides coupling to Gi, NMU2 also couples to Gq

It might be worth making the point that NMU2 appears to couple preferentially to Gq. Notably, not all authors have reported Gi-coupling, certainly from a functional perspective.

Line 295 As the C-terminal “wavy hook” of G - At the C-terminal “wavy hook” of G (?)

Lines 298-299 A highly conserved hydrophobic residues (I, L or F) - A highly conserved hydrophobic residue (I, L or F)

Lines 301-302 in a similar manner as previously reported. - in a similar manner to other GPCRs.

Lines 302-304 between NMU2 and Gi1 that intrigue NMU2 to engage G protein with this specific mode and reveals diversified Gi-coupled conformation – this sentence lacks clarity

Line 310 but the molecular mechanism is - but the molecular mechanisms are

Line 311 Here we determine the structure of NmU- - Here we determined the structure of the NmU-

Lines 312-313 depicting a comprehensive mechanism of NmU peptides in receptor recognition. - depicting a comprehensive mechanism of which NmU peptides are recognised by their cognate receptors.

Lines 314-315 most residues residing in the orthosteric binding pocket are conserved. - most residues residing in the orthosteric binding pocket are conserved.

Lines 315-316 we find out an incongruous pocket exist in the helices I and VII which effects the potency of – we demonstrate that an incongruous pocket exists in helices I and VII, which effects the potency of

Lines 317-320 Moreover, residues swapping in NMU2 with NMU1 of the incongruous pocket enhance the NMU2 response to selective agonist of NMU1. These evidences demonstrate the important role of these residues in ligand selectivity for NMUs. - Moreover, swapping residues of this pocket in NMU2 for those of NMU1 enhanced the NMU2 response to NMU1-selective agonists. This evidence demonstrates the important role of these residues in ligand selectivity for NMUs.

Lines 320-321 In addition, selective antagonist binding mode is illustrated by molecular docking. - In addition, a receptor-selective antagonist binding mode was illustrated by molecular docking.

Line 322 binding modes as well as selectivity mechanism - binding modes as well as a selectivity mechanism

Line 326 Structural comparison of NMU2 - Structural comparison of the NMU2

Line 327 represents a conserved activation process – suggests (or demonstrates) a conserved activation process

Line 329 in NMU2 - in the NMU2

Line 330 the dataset of NMU2 - the dataset of the NMU2

Line 331 of NMU2 - of the NMU2

Line 332 3D variance analysis did show - 3D variance analysis showed

Line 333 , which was similar to previous study on - , which was similar to a previous study on

Line 334 of α N - of the α N

Line 336 GPCR – GPCRs

Lines 337-339 All together, these findings would promote our understanding for the molecular mechanism of ligand recognition, selectivity and activation of NMU2, providing a reliable structural framework for rational drug design by targeting this receptor. - All together, these findings enhance our understanding of the molecular mechanisms of ligand recognition, selectivity and activation of NMU2, providing a reliable structural framework for rational drug design to target this receptor.

Accession codes are missing (lines 473 and 474) – noted that these will be added during revision.

'Span' in Suppl. Fig 1 would benefit from defining, particularly if this differs from Emax? SEM not needed in Suppl. Fig 1 as in legend.

Previous comment:

6. Although the determined structures and the bulk of the pharmacology (using the TRUPATH assay) have focused on Gi-coupling, the data around the antagonism of R-PSOP addresses IP accumulation and therefore Gq-coupling. Are there any issues around switching between different cellular outcomes with different transduction pathways?

This should be addressed in the MS.

8. For some of the mutants, there were significant effects on agonist potency in the TRUPATH assay. Again, is it possible that any of the shift in EC50 values were a consequence of alterations in expression? Further, where mutation reduced agonist potency without impacting receptor expression (e.g.H43E and N100A), would the data suggest that this may be a consequence of reduced ligand affinity or a reduction in ligand efficacy? Perhaps a little more discussion and clarity would add to the manuscript. --We thank the reviewer for this comment. The system we used for activity assays is overexpression system, and the NMU2 receptor showed a very high expression, and the actual expression level even for the mutants

with lower expression levels (i.e. ~30%) in comparison to the wild type receptor still relatively high. The activation curves of all the mutants are in a very good quality with good reproducibility, which were provided in the supplementary information. So we believe that the affinities determined of these mutant more likely reflect the real data. However, one could not rule out the possibility that the expression level also influence the affinity to some extent, so we have clarified this question by suggesting that the expression levels were also different and the affinity and the Emax measured might also affected by the lower level. For more detail please see response 7 to this Reviewer.

Just to comment that these comments again focus on ligand affinity. This has not been determined and the above argument should focus on ligand potency, with comment highlighting whether the structural evidence can be used to suggest changes in affinity (ligand binding) and/or receptor activation (conformational changes). Also just to note, whilst this is an over-expression system and 30% of WT expression may still be high, unless the receptor reserve for the measured response is known, this is not a valid argument; reductions of 70% may or may not impact on Emax but the evidence is not provided.

Point-by-Point response to reviewer comments

We appreciate all the reviewers for the constructive comments they provided. We have answered all questions in detail, and changes made in the manuscript are described below.

Reviewer #1 (Remarks to the Author):

Regarding my comments, the manuscript has been significantly improved and can be accepted in the current form.

--We appreciate the reviewer for the positive assessment about the manuscript.

Reviewer #2 (Remarks to the Author):

The revised manuscript by Zhao et al includes several modifications in response to this and other reviewers. An improved discussion, the addition of new data and a more thorough comparison with other GPCR-G protein structures improve the manuscript in general.

-- We are very grateful for the positive comments from the reviewer.

There are some remaining issues that need to be addressed:

Comments:

In their rebuttal the authors claim that the specific conformations of the lipid-bound versus detergent-stabilized NTS-G_i complexes are similar, implying that the lipids play little role. The authors should revisit this statement as careful examination of the two structures reveal significant differences. The most notable differences are between the receptor G protein interface, including the alphaN of Galphai and the interaction with ICL2. In lipid, ICL2 appears to be closer to the lipid face. This is likely influenced by the interaction of both ICL2 and the alphaN of Galphai with the phospholipid headgroups in the bilayer. In addition, the angle of attack of the C-term of Galphai is slightly different (approx. 12-15 degrees) in lipid, with respect to the receptor. In the detergent structure this rotation allows the alpha4-beta5 loop of Galphai to be closer to helix 8 of NTS1R, similar to what seems to be observed with NMUR1/2. Such an interaction is not possible in a phospholipid bilayer. So it is clear that the lipid plays a significant role in both receptor and G protein conformation.

--We thank the reviewer for this comment. We are totally agree with the reviewer about the significant role of the lipid in both receptor and G protein conformation. Lipid bilayer constrains the conformation of NTS1R to enhance its interaction with G_i and leads to the different interactions compared with the complex solved in detergent micelles. In the last rebuttal, we paid more attention to the rotation of the αN helix of Gα_{i1} and missed the different interaction modes between NTS1R and G_{i1} in lipid bilayer and detergent micelles. Represented as the relatively stable state of NMU2–G_{i1} complex,

the overall conformation between NMU2 and G_{i1} might not be changed much in lipid bilayer. But more structural information of NMU2–G_{i1} complex in lipid bilayer is needed to further explore the roles of lipid in NMU2 and G_{i1} conformation as well as detailed interaction patterns. To express the view more accurately, the statement “many structures of G protein-coupled GPCRs solved in detergent micelles or lipid bilayers have been obtained” in line 248-250 and “However, the NTS₁–G_i complex solved in lipid bilayers constrains the conformation of the receptor and shows the upward movement of helices VII and VIII, indicating the membrane association of this region. The specific interaction between helix VIII of the NMU2 and G_{αi1} might be affected in lipid bilayers and more structural information is need for further illustration about the roles of lipid in modulating the conformation of GPCR–G complexes.” in line 277-282 have been added for further explanation.

In line 51 the authors refer to two intrinsic GPCRs. Do the authors mean endogenous GPCRs or intrinsic membrane proteins or something else ?

--We are sorry for this misleading and the “intrinsic GPCRs” has been changed to “endogenous GPCRs”.

In the results section the authors should remind readers that they are using mutant version of Galphai. The mutant contains 5 mutations which effect nucleotide exchange, Mg2+ coordination and Switch II coordination (and therefore Gbetagamma interaction).

--We followed the reviewer’s suggestion and added the explanation of the mutant version of G_{αi1}. “Dominant-negative G_{αi1} (DNG_{αi1}) containing five mutations (S47C, G202T, G203A, E245A and A326S) was used to improve the stability of the complex as these mutations could lead to a preference for a nucleotide-free state and prevent the dissociation of Gβγ from the heterotrimer” in line 84-87.

In lines 170-172 the authors discuss E105(E2.64) and the decreased Emax. This should NOT be caused by the weakened affinity but could be affected by expression levels.

--We thank the reviewer for this comment. We changed the explanation about E105^{2.64} in line 174-175: “E105^{2.64}A also decreased the E_{max} of NmU-25 by 50%, which might be caused by reduction of the receptor expression level”.

In line 262-267 the authors make reference to an interaction between R333(R8.49) and the C-term of Galphai. This is unlikely to occur in a phospholipid bilayer since helix 8 would be buried in the bilayer, with the Arg side chains likely interacting directly with the phospholipid head group. The discussion of the positively-charged helix 8 following this sentence will likely have to be modified for this reason (see above discussion the lipid vs detergent structures).

--We agree with the reviewer’s comment. According to the first comment, we refined the point of view about the interaction between R333^{8.49} and the C-term of G_{αi}: “However, the NTS₁–G_i complex solved in lipid bilayers constrains the conformation of the receptor and shows the upward movement of helices VII and VIII, indicating the

membrane association of this region. The specific interaction between helix VIII of the NMU2 and G α_1 might be affected in lipid bilayers and more structural information is need for further illustration about the roles of lipid in modulating the conformation of GPCR–G complexes.” in line 277-282.

Minor Comments:

There are several grammatical errors (too many to list here) that need to be addressed. These are likely to be picked up by the editorial staff.

--We are sorry for these grammatical errors. According to the forth reviewer's suggestions, we have corrected grammatical errors carefully.

Line 186 'interacted' should read 'interacting'

--This error has been corrected.

Line 251 should read 'infected' instead of 'transfected'.

--The reviewer may refer the word “transfected” in line 360(?). We have changed “transfected” to “infected” as suggested.

Reviewer #3 (Remarks to the Author):

The authors have adequately addressed all of my comments, and the revised manuscript is substantially improved. The revised manuscript makes a valuable contribution to the field and I would recommend publication.

--We appreciate the reviewer for the positive assessment about the manuscript.

Reviewer #4 (Remarks to the Author):

My general comments about the paper made in the initial review still stand.

There are still a substantial number of typographical/grammatical errors that would, no doubt, be picked-up during the process.

Some suggested corrections/alterations for the Abstract, Introduction, Results and Discussion are given below. There are grammatical corrections required in other sections including the figure legends.

Below also contains one or two comments/questions.

--We sincerely appreciate the comments of the reviewer.

Line 36 high – highly

--The mistaken has been corrected.

Line 37 degree rotation of Gi protein in relevant to NMU2 compared with (?)

--For clarity, the sentence has been changed to “a 25-degree rotation of Gi protein compared with other solved Gi-bound complexes is also observed”.

Line 48 NmU-33 – NmS-33. Also, this line now perhaps provides misleading information about the distribution and roles of NmS-33. I would certainly argue that it is not ubiquitously expressed as indicated and may not play the full range of roles listed that NmU-25 may.

--We are sorry for the misleading information. After consulting more literatures, we have clarified information about NmS-33: “NmU-25 is ubiquitously distributed in the gastrointestinal tract, spinal cord and central nervous system. NmS-33 shows a more restricted distribution, being predominantly presented in the central nervous system. The system of NmU-25/NmS-33 implicates in the regulation of smooth muscle contraction, energy balance, feeding behavior, pronociception and tumorigenesis”.

Line 58 – the practical – practical

--This error has been corrected.

Line 59 NMUs – NMU

--This error has been corrected.

Line 63 activation of NMU2 would dramatically decrease body weight and food - activation of NMU2 dramatically decreases body weight and food

--This error has been corrected.

Line 64 intake in mice while in contrast inhibition of NMU2 would promote weight gain and - intake in mice while in contrast inhibition of NMU2 promotes weight gain and (?)

--This error has been corrected.

Line 66 is lagged behind partially due to the lack of structural information – has been limited, partially due to the lack of structural information.

--This error has been corrected.

Line 68 complex bound to endogenous peptide NmU - complex bound to the endogenous peptide NmU

--This error has been corrected.

Line 70 which is prerequisite for ligand recognition - which is a prerequisite for ligand recognition

--This error has been corrected.

Line 75-79 This lacks a little clarity. the flexible C-terminal residues Q356 to T415 of NMU2 were replaced by a twin-strep affinity tag and a hemagglutinin (HA) signal peptide followed by a flag tag was introduced at N terminus to improve expression

--For clarity, the sentence was rewritten: “To facilitate expression and purification of the NmU-25–NMU2–G_{i1} complex, the flexible C-terminal residues Q356 to T415 of the receptor were replaced by a PreScission protease site followed by a twin-strep affinity tag.”

Line 82 (named TRUPATH) – (TRUPATH)

--This error has been corrected.

Line 87 unambiguously with clear and strong density map (Supplementary Fig. 2g) - unambiguously with a clear and strong density map (Supplementary Fig. 2g).

--This error has been corrected.

Lines 88-90 The overall structure of NMU2 possesses canonical seven transmembrane helical domain compared with other solved peptide receptors of class A GPCRs with helix VIII unmodelled due to its flexibility upon activation - The overall structure of NMU2 possesses a canonical seven transmembrane helical domain similar to other solved peptide receptors of class A GPCRs with helix VIII unmodelled due to its flexibility upon activation (?)

--This error has been corrected as suggested.

Lines 94-95 The extracellular part of NMU2 is widely opened to accommodate peptide ligand is widely opened to accommodate peptide ligand compared with other solved peptide-bound receptors which share the high sequence similarity with NMU2 – To accommodate the peptide ligand, the extracellular part of NMU2 is more widely opened compared to other solved peptide-bound receptors. (?)

--This error has been corrected as suggested.

Lines 96-97 NMU2 adopts full active conformation in intracellular side stabilised - NMU2 adopts a fully active conformation at the intracellular side, stabilised

--This error has been corrected.

Line 101 indicate that NMU2 is in active state - indicate that NMU2 is in an active state

--This error has been corrected.

Line 102 is in active-like conformation - is in an active-like conformation

--This error has been corrected.

Line 103 the rearrangement of “P5.50-I3.40-F6.44” motif - the rearrangement of the P5.50-I3.40-F6.44 motif

--This error has been corrected.

Line 106 arousing the rearrangement of – causing the rearrangement of

--This error has been corrected.

Line 107 of $\alpha 5$ helix of G protein - of the $\alpha 5$ helix of the G protein

--This error has been corrected.

Line 108 to form a hydrogen network - to form a hydrogen network

--This error has been corrected.

Line 113 peptides with various forms of length share the - peptides with various lengths share the

--This error has been corrected.

Line 115 which is crucial for ligand's agonistic - which is crucial for the ligand's agonistic

--This error has been corrected.

Line 117 structure with the conserved C terminus penetrating deeply in the entire orthostatic - structure with the conserved C terminus, penetrating deeply in the entire orthostatic

--For clarity, the sentence has been rewritten: "NmU-25, the 25 amino acid endogenous agonist, forms a short β -meander structure and the conserved C terminus penetrates deeply in the entire orthostatic ligand binding pocket formed by ECL2, ECL3 and helices I-VII except for helix V of NMU2".

Line 121 of NMU2 plays minor role in ligand - of NMU2 plays a minor role in ligand

--This error has been corrected.

Line 123 contributes a large portion of interaction - contributes a large portion of the interaction

--This error has been corrected.

Line 126-127 to accommodate in the space between ECL1, ECL2 - to facilitate accommodation in the space between ECL1, ECL2

--This error has been corrected.

Line 128 For example, Y18 packs against with the main chain of ECL3 by hydrophobic interactions while - For example, Y18 packs against the main chain of ECL3 by hydrophobic interactions, while

--This error has been corrected.

Line 130 Besides, P199 - In addition, P199

--This error has been corrected.

Line 131 It has been known that NmS-33 showed a higher binding affinity for NMU2 overNmU-2528. This might be benefitted by potentially more interactions were formed between the hydrophilic residues such as arginine or glutamine in the extra N terminus

of NmS-33 and receptor ECL2, providing a strengthened interaction network between N terminus of peptide and the receptor. However, more data is needed before a clear conclusion was made. – There has been some indication that NmS-33 may have a higher binding affinity for NMU2 compared to NmU-2528. This might result from potentially more interactions between the hydrophilic residues such as arginine or glutamine in the extended N terminus of NmS-33 and receptor ECL2, providing a strengthened interaction network between the N terminus of the peptide and the receptor. However, more data are needed before a clear conclusion can be made.

--This error has been corrected.

Note that although this paper is often cited, there is little confirmatory data. In most studies, no real differences have been shown. Furthermore, the potencies of these two peptide ligands tend to be equivalent suggesting that unless NmS has a reduced intrinsic efficacy that precisely offsets a higher affinity, that this may not be the case.

--Followed the reviewer's suggestion, we have changed this citation to "Mori K, Miyazato M, Ida T, et al. Identification of neuromedin S and its possible role in the mammalian circadian oscillator system. EMBO J. 2005;24(2):325-335.". Competitive radioligand binding analysis in this paper demonstrated that human NmS has a higher binding affinity to NMU2 than did NmU ($IC_{50}=1.0\times 10^{-10}$ and 6.8×10^{-10} M, respectively).

Line 138 As for the C terminus, last four residues of NmU-25 – The last four C-terminal residues of NmU25

--This error has been corrected.

Line 143 caused a complete loss of the agonistic potency – have a complete loss of potency

--This error has been corrected.

Line 147 forms – form

--This error has been corrected.

Line 147 respectively and plays - respectively and play

--This error has been corrected.

Line 149 Similar recognition mechanism was also observed in monoamine receptors, which - Similar recognition mechanisms are also observed in monoamine receptors, which

--This error has been corrected.

Line 150 utilizing the acidic residues - utilize the acidic residues

--This error has been corrected.

Line 152-153 might play critical roles in mediating the stability of ligand-binding pocket which is - might play critical roles in mediating the stability of the ligand-

binding pocket, which is

--This error has been corrected.

Line 159 alanine impairs the receptor activation strikingly – strikingly impairs receptor activation

--This error has been corrected.

Line 160-162 These mutants also showed lower expression level in comparison of wild type receptor, which might also contribute to the weaker binding of corresponding ligands. - These mutants also showed lower expression levels in comparison of wild type receptor, which might also contribute to the weaker binding of corresponding ligands.

There needs to be some caution in such statements. Receptor-ligand affinity has not been measured. The authors have measured ligand potency. This is a reflection of affinity, efficacy and receptor expression. It is an assumption that it is affinity that is affected by mutagenesis. This may fit with the model but needs to be made clear (ie. that affinity has not been measured).

--We followed the reviewer's comment and rewritten the sentence: “Without ligand-binding analysis, the lower expression levels of these mutants, weaker binding affinity or reduced ligand efficacy of NmU-25 might impair the ligand-induced receptor activation.”

Line 164 alanine would result in inactive peptides - alanine results in inactive peptides

--This error has been corrected.

Lines 170-172 E1052.64A also decreased the E_{max} of NmU-25 by 50%, which might be caused by weakened binding affinity as well as the reduction of the receptor expression level (Supplementary Table 2).

A reduction in E_{max} is not consistent with a reduced binding affinity. Provided that the efficacy of a ligand is not reduced, a reduction in binding affinity would simply mean that a greater concentration of ligand would be required for the E_{max} – but it would be reached. A reduction in E_{max} can occur when efficacy is reduced (receptor mutation could cause this – the receptor is less well-coupled to signaling mechanisms) and/or receptor expression levels fall below those required for the maximal response (ie. the receptor reserve is removed).

--As the data shows that E105^{2.64}A decreased the E_{max} of NmU-25 by 50% and the expression level of E105^{2.64}A decreased by ~50% as well, we preferred to modify the sentence: “E105^{2.64}A also decreased the E_{max} of NmU-25 by 50%, which might be caused by reduction of the receptor expression level”.

Line 172-173 These structural evidences suggest – This structural evidence suggests

--This sentence refers not only to E105^{2.64}A, but also to the structural information

mentioned above. So “these structural evidences” might be better than “this structural evidence”.

--This error has been corrected.

Line 173 between this hydrophilic portion and the receptor - between this hydrophilic portion of the ligand and the receptor

--This error has been corrected.

Line 174 mediating agonistic functionality of NmU peptides - mediating agonism of NmU peptides

--This error has been corrected.

Line 175 alignment presents that these - alignment demonstrates that these

--This error has been corrected.

Line 178 pack against with each other - pack against each other

--This error has been corrected.

Line 183 (residue resides in NMU1) – (residue of NMU1) (?)

--This error has been corrected.

Line 184 which was further supported by weakening the potency of NmU-25 for - which was further supported by the reduced potency of NmU-25 for

--This error has been corrected.

Line 185-186 Residues interacted with - Residues that interact with

--This error has been corrected.

Line 187 MNU2 – NMU2

--This error has been corrected.

Line 190 CPN 116 and CPN 267 were - CPN 116 and CPN 267, were

--This error has been corrected.

Line 191 (residue 19-25) - (residues 19-25)

--This error has been corrected.

Line 191 (residue 19-25) with different selective characteristics - (residues 19-25). These had different selectivity characteristics

--This error has been corrected.

Line 193 CPN267 owned high agonistic activity for NMU1 - CPN267 showed high agonistic activity for NMU1

--This error has been corrected.

Line 196 position of 20 and 21. - position of residues 20 and 21.

--This error has been corrected.

Line 199 NMU2 complex reflects that residues interacted with L20 and F21 of - NMU2 complex indicates that residues that interact with L20 and F21 of

--This error has been corrected.

Line 201 residues affect the ligand selectivity - residues affect ligand selectivity

--This error has been corrected.

Line 202 NMU2 with relative residues in NMU1 - NMU2 with the respective residues of NMU1 (?)

--This error has been corrected.

Line 203 Substitution the H43 - Substitution of H43

--This error has been corrected.

Line 204 with bulky residues tryptophan - with the bulky tryptophan residue

--This error has been corrected.

Line 206-208 The side chain difference in the ligand binding pocket might serve to the designation of NMU1 and NMU2 selective ligands. - The side chain difference in the ligand binding pockets may underlie the receptor-selectivity of ligands. (?)

--This error has been corrected.

Line 209 glutamine which are the corresponding residue in NMU1 respectively – glutamine respectively, which are the corresponding residues of NMU1,

--This error has been corrected.

Line 210 caused about 40-fold reduction - caused about a 40-fold reduction

--This error has been corrected.

Lines 210-211 CPN 116 in stimulating NMU2 activation and A302 - CPN 116 in stimulating NMU2 signaling. A302

--This error has been corrected.

Lines 213-214 showed about 8-fold enhanced the potency compared with WT of CPN 267, a NMU1 selective ligand - showed about 8-fold enhanced potency with CPN 267, a NMU1 selective ligand, compared to the WT receptor

--This error has been corrected.

Line 215 In the meanwhile, - In addition,

--This error has been corrected.

Line 216 had no improvement at all – had no effect

--This error has been corrected.

Line 217 These results suggested that - These results suggest that

--This error has been corrected.

Lines 218-219 and the results will provide the structural basis for future designation of selective drugs. - and provide a structural basis for the future design of receptor-selective drugs.

--This error has been corrected.

Line 220 binding mode of non-peptide antagonist - binding mode of a non-peptide antagonist

--This error has been corrected.

Line 225 stabilized through the polar interactions - stabilized through polar interactions

--This error has been corrected.

Lines 226-227 are bordered by hydrophobic - are bordered by a hydrophobic

--This error has been corrected.

Line 227 F2846.51, F3167.42 - F2846.51 and F3167.42

--This error has been corrected.

Line 231 we carried out functional experiment by an inositol phosphate (IP) accumulation assay to test the effects of these residues involved in R-PSOP binding. - we carried out an inositol phosphate (IP) accumulation assay to test the effects of these residues involved in R-PSOP binding.

--The above issues have been corrected.

It might be worth highlighting that this assay is likely to measure predominantly Gq-coupling (there is no evidence of eg. pertussis toxin-sensitive (Gi-mediated) inositol phosphate accumulation). How does this relate to the structural data on Gi-coupling?

--We thank the reviewer for this comment. Actually, we only utilized the model of NMU2 rather than the NMU2–G_{i1} complex for molecular docking. The software predicts and identifies the most favorable binding mode of a given ligand in the binding pocket of a given receptor. So the result of molecular docking is not related to the structural data on G_i-coupling, it's only related to the structural data of the NMU2. We think that an IP assay is feasible to test the antagonism of R-PSOP.

Lines 235-237 which might cause by lowering the expression level of the receptor greatly (Supplementary Fig. 7e, Supplementary Table 3). - which may be a consequence

of the dramatically reduced expression level (Supplementary Fig. 7e, Supplementary Table 3).

--This error has been corrected.

Line 239 V is not conserved between NMU1 and NMU2 - V are not conserved between NMU1 and NMU2

--This error has been corrected.

Line 242 with leucine or cysteine could decrease the antagonistic - with leucine or cysteine decreased the antagonistic

--The above issues have been corrected.

Line 245 ligand selectivity for non-peptide antagonist. - ligand selectivity for the non-peptide antagonist. or ligand selectivity for non-peptide antagonists. (depending on whether this is a specific statement relating to R-PSOP or non-peptide antagonists generally).

--Since only one non-peptide antagonist of NMU2 has been published so far, we prefer the first statement.

Line 251 in NMU2–Gi1 - in the NMU2–Gi1

--This error has been corrected.

Line 251 to that in hNTS - to that in the hNTS

--This error has been corrected.

Lines 252-253 (Fig. 4a). $\alpha 5$ helix is tilted with different degrees compared with Gai1 in other GPCR–Gi complex (Fig. 4b). - $\alpha 5$ helix is tilted to a different extent compared with Gai1 in other GPCR–Gi complexes (Fig. 4b).

--This error has been corrected.

Line 253 Similar conformational difference in G – A similar conformational difference in G

--This error has been corrected.

Line 256 induced by the receptor itself and NMU2 - induced by the receptor itself and that the NMU2

--This error has been corrected.

Line 262 and NMU2 compared to other Gai1 complexes - and NMU2 compared to other GPCR-Gai1 complexes (?)

--This error has been corrected.

Line 265 forms a salt bridge with C-terminal - forming a salt bridge with the C-terminal

--This error has been corrected.

Line 266 there is no other - there are no other

--This error has been corrected.

Line 268 conserved hydrogen bond between C-terminal carboxyl of Ga - conserved hydrogen bond between the C-terminal carboxyl of Ga

--This error has been corrected.

Lines 270-271 coupled with Gi/o protein reveals that only a small number of receptors have a positively - coupled with the Gi/o protein reveals that only a small number of receptors have positively

--This error has been corrected.

Line 274 with G protein - with the G protein

--This error has been corrected.

Line 279 ICL2 played - ICL2 plays

--This error has been corrected.

Line 281 receptor own a bulky - receptor have a bulky

--This error has been corrected.

Line 284 of α N helix - of the α N helix

--This error has been corrected.

Line 285 of receptor and G - of the receptor and G

--This error has been corrected.

Line 286 Similarly, the hNTS - Similarly, hNTS

--This error has been corrected.

Lines 286-287 it also mediate weaker - it also mediates weaker

--This error has been corrected.

Line 287 of hNTS - of the hNTS

--This error has been corrected.

Line 288 compared with β 2AR - compared with the β 2AR

--This error has been corrected.

Lines 290-291 even though with same aromatic amino acid present - even though the same aromatic amino acid is present

--The above issues have been corrected.

Lines 292-293 Besides Gi signaling pathway, NMU2 was reported to couple Gq for signaling 292 transduction as well - Besides coupling to Gi, NMU2 also couples to Gq. It might be worth making the point that NMU2 appears to couple preferentially to Gq. Notably, not all authors have reported Gi-coupling, certainly from a functional perspective.

--We can't explain the reason why not all authors have reported Gi-coupling of the NMU2. But at least two researches provided robust experimental data and verified that NMU2 couples to Gi with nano-molar potency(Aiyar, Disa et al. 2004, Brighton, Szekeres et al. 2004). And the TRUPATH data represented in the last response could approve it as well.

Line 295 As the C-terminal "wavy hook" of G - At the C-terminal "wavy hook" of G (?)

--This error has been corrected.

Lines 298-299 A highly conserved hydrophobic residues (I, L or F) - A highly conserved hydrophobic residue (I, L or F)

--This error has been corrected.

Lines 301-302 in a similar manner as previously reported. - in a similar manner to other GPCRs.

--The above issues have been corrected.

Lines 302-304 between NMU2 and Gi1 that intrigue NMU2 to engage G protein with this specific mode and reveals diversified Gi-coupled conformation – this sentence lacks clarity

--For clarify, the sentence has been rewritten: "the specific interactions between the NMU2 and Gi1 intrigue the NMU2 to engage G protein with the specific conformational state, revealing diversified GPCR-Gi coupling mode. This will provide more structural basis for understanding the complexity of GPCRs and Gi protein coupling.".

Line 310 but the molecular mechanism is - but the molecular mechanisms are

--This error has been corrected.

Line 311 Here we determine the structure of NmU- - Here we determined the structure of the NmU-

--This error has been corrected.

Lines 312-313 depicting a comprehensive mechanism of NmU peptides in receptor recognition. - depicting a comprehensive mechanism of which NmU peptides are recognised by their cognate receptors.

--This error has been corrected.

Lines 314-315 most residues resided in the orthosteric binding pocket are conserved. -

most residues residing in the orthosteric binding pocket are conserved.

--This error has been corrected.

Lines 315-316 we find out an incongruous pocket exist in the helices I and VII which effects the potency of – we demonstrate that an incongruous pocket exists in helices I and VII, which effects the potency of

--This error has been corrected.

Lines 317-320 Moreover, residues swapping in NMU2 with NMU1 of the incongruous pocket enhance the NMU2 response to selective agonist of NMU1. These evidences demonstrate the important role of these residues in ligand selectivity for NMUs. - Moreover, swapping residues of this pocket in NMU2 for those of NMU1 enhanced the NMU2 response to NMU1-selective agonists. This evidence demonstrates the important role of these residues in ligand selectivity for NMUs.

--This error has been corrected.

Lines 320-321 In addition, selective antagonist binding mode is illustrated by molecular docking. - In addition, a receptor-selective antagonist binding mode was illustrated by molecular docking.

--This error has been corrected.

Line 322 binding modes as well as selectivity mechanism - binding modes as well as a selectivity mechanism

--This error has been corrected.

Line 326 Structural comparison of NMU2 - Structural comparison of the NMU2

--This error has been corrected.

Line 327 represents a conserved activation process – suggests (or demonstrates) a conserved activation process

--This error has been corrected.

Line 329 in NMU2 - in the NMU2

--This error has been corrected.

Line 330 the dataset of NMU2 - the dataset of the NMU2

--This error has been corrected.

Line 331 of NMU2 - of the NMU2

--This error has been corrected.

Line 332 3D variance analysis did show - 3D variance analysis showed

--This error has been corrected.

Line 333 , which was similar to previous study on - , which was similar to a previous study on

--This error has been corrected.

Line 334 of αN - of the αN

--This error has been corrected.

Line 336 GPCR – GPCRs

--This error has been corrected.

Lines 337-339 All together, these findings would promote our understanding for the molecular mechanism of ligand recognition, selectivity and activation of NMU2, providing a reliable structural framework for rational drug design by targeting this receptor. - All together, these findings enhance our understanding of the molecular mechanisms of ligand recognition, selectivity and activation of NMU2, providing a reliable structural framework for rational drug design to target this receptor.

Accession codes are missing (lines 473 and 474) – noted that these will be added during revision.

--The accession codes (PDB 7XK8, EMD-33247) have been added in line 480-481.

‘Span’ in Suppl. Fig 1 would benefit from defining, particularly if this differs from E_{max} ? SEM not needed in Suppl. Fig 1 as in legend.

--We have added the explanation about span: “The span is defined as the window between the maximal agonist response (E_{max}) and minimal agonist response.”. SEM has been occluded as reviewer suggested.

Previous comment:

6. Although the determined structures and the bulk of the pharmacology (using the TRUPATH assay) have focused on G_i -coupling, the data around the antagonism of R-PSOP addresses IP accumulation and therefore G_q -coupling. Are there any issues around switching between different cellular outcomes with different transduction pathways?

--As the answer to the above comment, we use IP accumulation to test the antagonism of R-PSOP for two reasons: the first point is that the molecular docking of R-PSOP only used the model of NMU2, the result of molecular docking is not related with the G_i -coupling; the second point is that we used the EC_{50} ratio ($EC_{50(NmU-25 + R-PSOP)}/EC_{50(NmU-25)}$) to assess the antagonistic efficacy of R-PSOP. The result represents the antagonistic efficacy of R-PSOP with regardless of G_i or G_q transduction pathway.